🔓 | **Open Peer Review** | Eukaryotic Cells | Research Article

# With a little help from my friends: importance of protist-protist interactions in structuring marine protistan communities in the San Pedro Channel

Samantha J. Gleich,[1] Lisa Y. Mesrop,[2] Jacob A. Cram,[3] J. L. Weissman,[4,5] Sarah K. Hu,[6] Yi-Chun Yeh,[1,7] Jed A. Fuhrman,[1] David A. Caron[1]

**ABSTRACT** Marine protists form complex communities that are shaped by environmental and biological ecosystem properties, as well as ecological interactions between organisms. While all of these factors play a role in shaping protistan communities, the specific ways in which these properties and interactions influence protistan communities remain poorly understood. Fourteen years and 9 months of eukaryotic amplicon (18S-V4 rRNA gene) data collected monthly at the San Pedro Ocean Time-series (SPOT) station were used to evaluate the impacts that environmental and biological factors, and protist-protist interactions had on protistan community composition. Statistical analysis of the amplicon data revealed that seasonal patterns in protistan community composition were apparent, but that the environmental data collected through routine time-series sampling efforts could not explain most of the variability that was evident in the communities. To identify some of the protist-protist interactions that may have played a role in shaping protistan communities, ecological networks were constructed using the amplicon data and the network predictions were compared against a database of confirmed protist-protist interactions. The database comparisons revealed hundreds of established parasitic, predator-prey, photosymbiotic, and mutualistic relationships in the networks. Although many interactions were confirmed using the database, these confirmed interactions constituted only 2% of the interactions identified at the SPOT station, highlighting the need to better characterize protist-protist interactions in marine environments. Finally, the network-predicted interactions that were not found in the database were used to identify putative, novel protist-protist interactions that may have played a role in structuring the protistan communities at the SPOT station.

**IMPORTANCE** Network analyses are commonly used to identify some of the ecological interactions that may be occurring between protists in the ocean; however, evaluating predictions obtained from these analyses remains difficult due to the large number of interactions that may be recovered and the limited amount of information available on protist-protist interactions in nature. In this study, ecological network analyses were conducted using data collected at the San Pedro Ocean Time-series (SPOT) station and the network predictions were compared against a database of established protist-protist interactions. These database comparisons revealed hundreds of confirmed protist-protist interactions, and thousands of putative, novel interactions that may be occurring at the SPOT station. The database comparisons carried out in this study provide a new way of evaluating network predictions and highlight the complex, yet critical role that ecological interactions play in shaping protistan community composition in marine ecosystems.

**Peer Reviewer** Sean Anderson, University of New Hampshire, Durham, New Hampshire, USA

Address correspondence to Samantha J. Gleich, gleich@usc.edu.

The authors declare no conflict of interest.

See the funding table on p. 19.

**KEYWORDS** cell-cell interaction, environmental microbiology, protists, microbial communities

Understanding the ways in which biotic and abiotic factors impact the diversity of microbial eukaryotes (i.e., protists) has been a central focus in marine microbial ecology for decades (1). The pivotal roles played by abiotic variables such as temperature, light, and nutrients in structuring protistan communities have been highlighted through numerous global sampling efforts (2–5). Additionally, time-series studies have been used to identify seasonal, annual, and interannual patterns in protistan community composition that coincide with changes in the physiochemical environment (6–10). While these studies have established that environmental variables play an important role in influencing protistan community structure in the ocean, they have also revealed that these factors alone cannot completely explain the variability that has been observed in marine protistan communities across time and space (4, 7, 9, 11).

Microbe-microbe interactions provide another explanation for the spatial and temporal variability observed in protistan community composition. Biotic interactions manifest themselves in many trophic relationships including mutualism, parasitism, commensalism, and predation (12, 13). Moreover, biotic interactions can occur between protists and other protists, protists and prokaryotes (i.e., bacteria and archaea), protists and viruses, and protists and metazoans (11, 14). Studies carried out in aquatic ecosystems in recent years have highlighted that these biotic interactions play critical roles in structuring protistan communities, implying that research to delineate species-species associations will improve our understanding of the factors controlling protistan community structure (9, 11, 15–20). Our limited knowledge of planktonic, species-species interactions is in part due to the large diversity of microorganisms that exist in aquatic environments (21, 22), the multitude of interaction types that occur in nature (23, 24), and the difficulties associated with studying interactions that are taking place at relatively short timescales (25–27). However, high-throughput sequencing technologies and modern statistical approaches are now enabling researchers to explore these interactions (28).

Ecological network analyses are useful tools that can identify putative microbial interactions in nature. Species relative abundance data (inferred from relative sequence abundances) can be used in network analyses to determine how organismal abundances change in relation to one another, enabling the identification of potential organismal interactions (25, 29, 30). Results obtained from network analyses have been used to formulate hypotheses regarding specific biotic interactions that may occur in a system, and the role(s) that these biotic interactions play in shaping microbial community dynamics (31–36). However, evaluating the predictions obtained from network analyses remains difficult because most of the predicted interactions have not been confirmed (27). Moreover, the co-occurrence patterns detected through network analyses do not provide information on the types of interactions that are occurring (e.g., parasitism and mutualism), although the type of interaction can sometimes be inferred using known information about the organisms involved. One way to address these challenges is to compare network-derived associations to a list of known interactions that have been confirmed in culture, controlled experiments, or field observations. This type of informed approach can verify specific network predictions, and draw attention to putative, novel associations that may be occurring between microorganisms in the natural environment (11, 13).

In this study, we collected microbial eukaryotic amplicon sequence data at the San Pedro Ocean Time-series (SPOT) station over 14 years and 9 months to investigate the effects that biotic and abiotic factors have on protistan community composition and structure. We used Mantel tests, Bray-Curtis similarity metrics, and constrained ordination analyses to identify significant temporal patterns in protistan community composition and how these patterns related to concomitant physiochemical and biological measurements. Then, we carried out network analyses to document putative, protist-protist

associations that may have played a role in structuring the SPOT protistan communities. We compared these putative associations against a database containing established protist-protist interactions (Protist Interaction Database [PIDA] [13]) to (i) determine whether the network analyses could capture established protist-protist interactions, and (ii) identify the types of protist-protist interactions that may have played a role in shaping protistan community dynamics at the SPOT station.

## MATERIALS AND METHODS

### Field site

The SPOT station is located in the Southern California Bight, approximately halfway between Santa Catalina Island and the Palos Verdes Peninsula, California (33°33′N, 118°24′W; Fig. S1). The physicochemical properties of the waters at the SPOT station and the microbial communities present in these waters have been studied monthly since 1998 as part of a time-series program (https://dornsife.usc.edu/spot/). Through these time-series sampling efforts, shifts in protistan community composition have been observed over time, space, and depth using microscopy (37) and various DNA-based approaches including 18S rRNA gene clone libraries (38, 39), terminal restriction fragment length polymorphism (T-RFLP) (6, 40), and high-throughput sequencing technologies (9, 10, 41, 42). Samples have been collected at the SPOT station monthly since 2003 to characterize protistan community composition using amplicon sequencing of the V4 hypervariable region of the 18S rRNA gene (43). A small subset of these 18S-V4 amplicon data was used in a 1-year (i.e., 2013–2014) analysis of protistan community composition and activity at the SPOT station by Hu et al. (41). The other 13 years of 18S-V4 data that were collected during this time-series have not been analyzed or discussed in previous SPOT publications.

### Sample and data collection

Samples analyzed in this study were collected monthly from September 2003 through June 2018 at 5 m depth (herein referred to as "surface") and the deep chlorophyll maximum (herein referred to as "DCM"; Table S1). There were 56 months missed at the surface and 58 months missed at the DCM during the 14-year and 9-month long time-series sampling period due to equipment or weather-related issues. In total, 122 surface water samples and 120 DCM samples were collected and processed without replicates. Seawater samples were collected using a Niskin rosette, and prefiltered sequentially through 200 µm and 80 µm Nitex mesh to remove most metazoan zooplankton. Aliquots (2 L) of the prefiltered seawater were filtered onto a 0.7 µm GF/F filter (Whatman, International Ltd., Florham Park, NJ, USA) from each depth. The filters were placed in 2 mL of lysis buffer (100 mM Tris [pH = 8], 40 mM EDTA, 100 mM NaCl, and 1% SDS), flash-frozen in liquid nitrogen, and stored at −80°C until processed.

A variety of environmental and biological factors (referred to as environmental/biological parameters herein) were collected as part of the time series and were used in downstream analyses (Table S2). The environmental factors included physiochemical and temporal variables such as seawater temperature, salinity, and the day of the year, while the biological factors included variables influenced by organismal abundances or physiology such as chlorophyll *a* fluorescence, primary production rate estimates, and the relative abundances of specific prokaryotic taxa (Table S2). CTD measurements (SeaBird, Bellevue, WA, USA) were recorded at the time and depth of sample collection (temperature, beam attenuation, chlorophyll *a* fluorescence, dissolved oxygen, and salinity). Water samples were collected for nutrient analyses and measurements of dissolved oxygen concentrations via Winkler titration (44). Ammonium, nitrate + nitrite, silica, and phosphate analyses were carried out by the Marine Science Institute Analytical Lab at the University of California Santa Barbara. Satellite-derived sea surface height measurements were obtained from the Copernicus Marine

Service website (https://www.copernicus.eu/en) and averaged monthly, satellite-derived chlorophyll a concentrations, sea surface temperatures, and primary production rates at the SPOT station were obtained from the NOAA Coastwatch website (https://coast-watch.pfeg.noaa.gov/erddap/index.html). The multivariate ENSO index (MEI) values associated with each monthly sampling effort were also obtained from the NOAA website. Total relative abundances of free-living (0.2–1 µm) and particle-associated (1–80 µm) Flavobacteriales, SAR 11, and Synechococcales were obtained using 16S rRNA gene amplicon data collected by Yeh and Fuhrman (10) in a previous SPOT publication. Relative abundances of Flavobacteriales and the SAR11 clade were used in the current study because these two bacterial orders had some of the highest average prokaryotic abundances at the SPOT station over a 14-year period (10). The Synechococcales were used because of the important role that members of this order (e.g., *Prochlorococcus* and *Synechococcus*) play in primary production in aquatic environments. Relative abundances of ASVs that were assigned to each of these orders were summed for each sample across the data set. All environmental/biological data points that were missing in the time-series data set were imputed using the missForest function in the missForest package in R (45).

## DNA extraction, library preparation, sequencing, and bioinformatic analyses

Total DNA was extracted from the filters using the Qiagen DNA/RNA AllPrep kit (Qiagen, Valencia, CA, USA, #80204). A two-step library preparation procedure was used to amplify the V4 hypervariable region of the 18S rRNA gene (18S-V4). The 18S-V4 library preparation was carried out using the Stoeck et al. (43) primers (forward: 5′-CCAGCASCYGCGG-TAATTCC-3′; reverse: 5′-ACTTTCGTTCTTGATYRA-3′) and Illumina P5 and P7 indices as described by Hu et al. (46). PCR products were purified using Agencourt AMPure XP beads (Beckman Coulter, Brea, CA, USA, #A63881) and quantified using a Qubit 2.0 (ThermoFisher Scientific, Waltham, MA, USA, #Q32866). Samples were normalized and pooled following PCR amplification, indexing and purification. The pooled 18S-V4 rRNA gene library was sequenced on a MiSeq 2 × 250 paired-end sequencing platform at Laragen (Culver City, CA, USA). Laragen provided demultiplexed FASTQ files that were used in the downstream bioinformatic analysis.

DADA2 was used to create amplicon sequence variants (ASVs) from the sequence data within the QIIME2 (v.2022.2) software package (47–49). Raw sequences were first trimmed to remove Illumina adapters and base pairs with a quality score below 25 across a 10-base-pair sliding window using Trimmomatic (v.0.38) (50). Cutadapt (v.4.0) (51) was then used within the QIIME2 environment to remove the 18S-V4 primers. Forward and reverse reads were truncated at 210 and 200 base pairs, respectively, using DADA2 to remove additional low-quality base pairs and ASVs were generated from the truncated sequences using the pooled chimera detection method in DADA2. An 18S-V4 classifier was created in the QIIME2 software package using the PR2 database (v.5.0.0) (52). Taxonomy was assigned to the ASVs using this classifier and the QIIME2 classify-sklearn method with default settings.

## Analysis of protistan community composition over time

All ASVs were used to identify seasonal, annual, and interannual patterns in protistan community composition. First, the sum of the relative abundances of ASVs associated with major protistan taxonomic groups was calculated and averaged for each month at the two sampling depths (surface and DCM). Relative abundances of the ASVs were then used to calculate the Bray-Curtis dissimilarity between every pair of samples using the vegan package in R (53, 54) and these Bray-Curtis dissimilarity values were subtracted from one to calculate Bray-Curtis similarity. The number of days between samples was calculated and the number of months between samples was obtained by dividing the number of days by the average length of a month (30.4 days). The mean Bray-Curtis similarity between pairs of samples collected 1–177 months apart and the standard errors associated with these values were calculated. These mean Bray-Curtis

similarity values were plotted as a function of the number of months between samples to visualize how similar protistan communities were to one another seasonally, annually, and interannually at each depth. Finally, Mantel tests were carried out using Spearman correlations in the ape package in R (54, 55) to determine if the time (in days) between samples was significantly correlated with the Bray-Curtis dissimilarity between samples.

After identifying temporal patterns in protistan community composition, an analysis was performed to determine whether changes in protistan community composition over time were associated with changes in the environmental and biological ecosystem properties that were collected as part of routine time-series sampling efforts. The environmental and biological data were standardized so that the variables had a mean value of 0 and a variance of 1 using the decostand function in the vegan package in R (53, 54). These standardized values were then used to carry out a principal component analysis (PCA) at each depth using the vegan package (53, 54). The first principal component (that explained the largest fraction of the variability at each depth) was then extracted and the Euclidean distance of this principal component was calculated between all pairs of samples at each depth. These Euclidean distances were plotted as a function of the Bray-Curtis similarity values that were previously calculated for each pair of samples, and a simple linear regression was used to evaluate the significance of the relationship between the Bray-Curtis similarity values and the Euclidean principal component distances.

Redundancy analyses (RDAs) were performed to determine what fraction of the variability in protistan community composition could be explained by the environmental/biological parameters that were measured throughout the time series. The ASV abundance data were first normalized using a centered log-ratio (CLR) transformation in the compositions package in R to account for the compositional nature of the abundance data (54, 56). The environmental/biological data used to constrain the RDA axes were then standardized so that the variables had a mean value of 0 and a variance of 1 using the decostand function in the vegan package in R (53, 54). The normalized ASV abundance data were first used to parameterize empty models (with no environmental/biological predictors) and global models (with all environmental/biological predictors; Table S2) at the surface and DCM. Then, the empty models were fit using a forward selection procedure with the ordiR2step function in the vegan package in R (53, 54) to identify which of the 24 environmental/biological variables in the global models were significantly associated with protistan community composition at each depth. Environmental/biological variables were considered significant in driving compositional differences across samples if the Bonferroni-adjusted $P$ value resulting from the forward selection procedure was less than 0.05 for a given variable. The variables that were statistically significant at each depth were used to fit a final RDA model at the surface and DCM.

## Network inference and evaluation

Network analyses were carried out using both ASVs and environmental/biological parameters (Table S2) as input. The ASVs used in the networks were present (i.e., non-zero) in at least 20% of the surface or DCM samples. This 20% threshold was selected in an effort to include as many ASVs as possible in the analysis, while ensuring that the networking procedure remained computationally feasible. In total, there were 1,326 ASVs that were used in the network analyses (≈8% of the surface and DCM communities). The abundances of the ASVs that were used in the network analyses were normalized using a CLR transformation (compositions package in R [54, 56]) to account for the compositional nature of the ASV abundance data. The normalized ASV abundances and the environmental/biological parameters (Table S2) were then transformed using the NetGAM package in R to decrease the influence that temporal features had on the ASV abundance data and environmental/biological measurements prior to network construction (54, 57). A nonparanormal transformation was applied to the NetGAM-transformed data using the huge.npn function in the huge package in R

(54, 58) to ensure that the distribution of the transformed data resembled a multivariate normal distribution. Surface and DCM networks were constructed separately using a graphical lasso (glasso) networking approach (59). The glasso networks use a conditional independence approach to decrease the likelihood of capturing associations that result from indirect interactions, yielding a sparser network output than correlation-based approaches (60). Moreover, the glasso network uses a stability selection technique to identify the most stable network output, but does not provide the $P$ values or correlation coefficients that would be obtained using a correlation-based networking approach. The glasso networks used in this study were constructed by generating a list of 30 regularization parameter values (i.e., lambdas) and implementing the pulsar function in R (pulsar package in R; rep.num = 50, criterion = "stars"; [54, 61, 62]). The final surface and DCM networks were merged using the igraph package in R (54, 63) to identify associations that were detected at both depths.

The ASV-ASV associations that were identified in the network outputs were cross-checked against the PIDA (13), which contains ~2,500 established protist-protist, protist-bacteria, and protist-archaea interactions. The interactions in PIDA are classified into four interaction types: predation, parasitism, symbiosis (i.e., mutualism), and "unresolved interactions." The "unresolved interactions" documented in PIDA are interactions where it is unknown if the interaction is beneficial or neutral to the organisms involved (13) and were combined with the "symbiosis" category in our analysis (referred to as "other symbioses" in the downstream analyses). Predation was the most common protist-protist interaction type recorded in PIDA (~43% of all protist-protist interactions), followed by "other symbioses" (~36% of all protist-protist interactions) and parasitism (~22% of all protist-protist interactions).

When a SPOT network association matched a relationship documented in the PIDA database, we considered the relationship to be "PIDA-supported." Genus- and species-level taxonomic identifications were not always obtained for ASVs, especially when ASVs were affiliated with taxonomic groups that have been historically understudied (e.g., Syndiniales and MAST). Conversely, studies that describe organismal associations (such as those cited in PIDA) almost always characterize associations at the genus or species level. To address the discrepancies between ASV taxonomy and PIDA taxonomy, Syndiniales parasitic relationships were considered supported by the PIDA database when the Syndiniales group and host genus in the PIDA database matched a network prediction. Similarly, acantharean and polycystine photosymbioses were identified as PIDA-supported when an acantharean (Clade F) or polycystine host was associated with a known, symbiont genus. Symbiotic relationships containing MAST cells were considered PIDA-supported when the MAST clade and the genus of the symbiont matched an association documented in the PIDA database. All other network-derived interactions that were considered PIDA-supported matched the associations documented in PIDA at the genus level. The PIDA-supported interactions were plotted at the surface and DCM as subnetworks (i.e., subsets of the full network) to visualize the protist-protist relationships that have been established in other studies and may be occurring *in situ* at the SPOT station.

The network-predicted interactions that were not present in the PIDA database were used to identify several putative, novel interactions. The ASV abundance data were CLR-transformed (54, 56), and the Spearman correlation coefficient (SCC) was calculated between every pair of ASVs that were involved in a network-predicted interaction. Some of the strongest (i.e., highest absolute value SCC) network-predicted interactions that were not found in the PIDA database were visualized to identify some of the potentially novel protist-protist interactions that may be occurring *in situ* at the SPOT station.

## RESULTS AND DISCUSSION

In this study, we sought to determine the role(s) that specific environmental/biological parameters (Table S2) and protist-protist interactions played in structuring protistan community composition at the SPOT station by leveraging a long-term (14-year and

9-month) time-series data set. We were able to differentiate previously established interactions in our networks from those that may represent undiscovered associations by comparing our network predictions to a database of established protist-protist interactions. We were also able to characterize some of the types of interactions that may be occurring between protists at the SPOT station (e.g., parasitism and predation) using these database comparisons. The information gleaned from our analyses revealed that the ecological interactions occurring between protists at the SPOT station are diverse and critically important in influencing community dynamics in this system.

## Similarities and differences in surface and DCM protist communities

The 14-year and 9-month long SPOT time-series data set yielded a total of 29,123 eukaryotic ASVs across 122 surface samples and 120 DCM samples. The composition of the surface and DCM communities were similar to one another and fairly stable seasonally when examined at the level of broad taxonomic groupings and averaged monthly over the sampling period (Fig. 1). These findings were in general agreement with previous studies that have documented high relative abundances of dinoflagellates and Syndiniales throughout the year at the SPOT station using other molecular methods such as 18S rRNA gene clone libraries (38, 39), T-RFLP (6), and 515Y-926R amplicon sequencing (10). Subtle differences between the surface and DCM samples were apparent when examined at the level of ASV diversity (Fig. S2). Analysis of similarities (ANOSIM) on the Bray-Curtis dissimilarity matrix of the surface and DCM communities confirmed that the communities at these two sampling depths were significantly different from one another ($P < 0.01$; $R = 0.32$) despite some similarities in community composition that were observed between the depths (Fig. S2).

## Temporal variability and cyclicity in protistan community composition

Temporal patterns in protistan community composition were observed throughout the study period at the SPOT station. The average Bray-Curtis similarity values calculated between pairs of samples collected across different time intervals revealed that protistan communities observed 12 months apart at the SPOT station were more similar to one another than communities that were observed 6 months apart (Fig. 2). The lower average Bray-Curtis similarity values calculated for samples collected at 6-month intervals (e.g., 6, 18, and 30 months; Fig. 2) relative to 12-month intervals (e.g., 12, 24, and 36 months; Fig. 2) highlight the cyclical, seasonal patterns in protistan community composition at the SPOT station. Moreover, these Bray-Curtis similarity calculations confirm that repeatable, seasonal dynamics are prevalent over decadal scales at the SPOT station.

The average Bray-Curtis similarity values calculated across different time intervals also indicated that the protistan communities changed slightly across a 10-year study period (i.e., general decreases in the average Bray-Curtis similarity with increases in the number of months between samples up to approximately 120 months; Fig. 2). However, decreases in community similarity through time were extremely variable. For example, at both the surface and DCM, there were linear decreases in community similarity over time for the first four to five sampling years (i.e., general decreases in the average Bray-Curtis similarity from 1 to 48 months at the surface and 1 to 60 months at the DCM; Fig. 2). Then, the average Bray-Curtis similarity values stabilized over a 3- to 4-year period before declining again up to approximately 120 months (Fig. 2). These decreases in community similarity were quantitatively assessed using Mantel tests for the surface and DCM samples and revealed that communities became significantly more dissimilar over the entire study period at both depths ($P < 0.01$). Moreover, the Bray-Curtis similarity between samples was plotted as a function of the Euclidean distance of the environmental/biological ecosystem measurements between samples (derived from the first principal component of a PCA; Fig. S3). This analysis revealed that protistan communities were generally less similar when large differences in environmental/biological properties were observed between samples (i.e., negative slopes in Fig. S3), and this trend was more pronounced in surface waters. The seasonal and interannual shifts in protistan

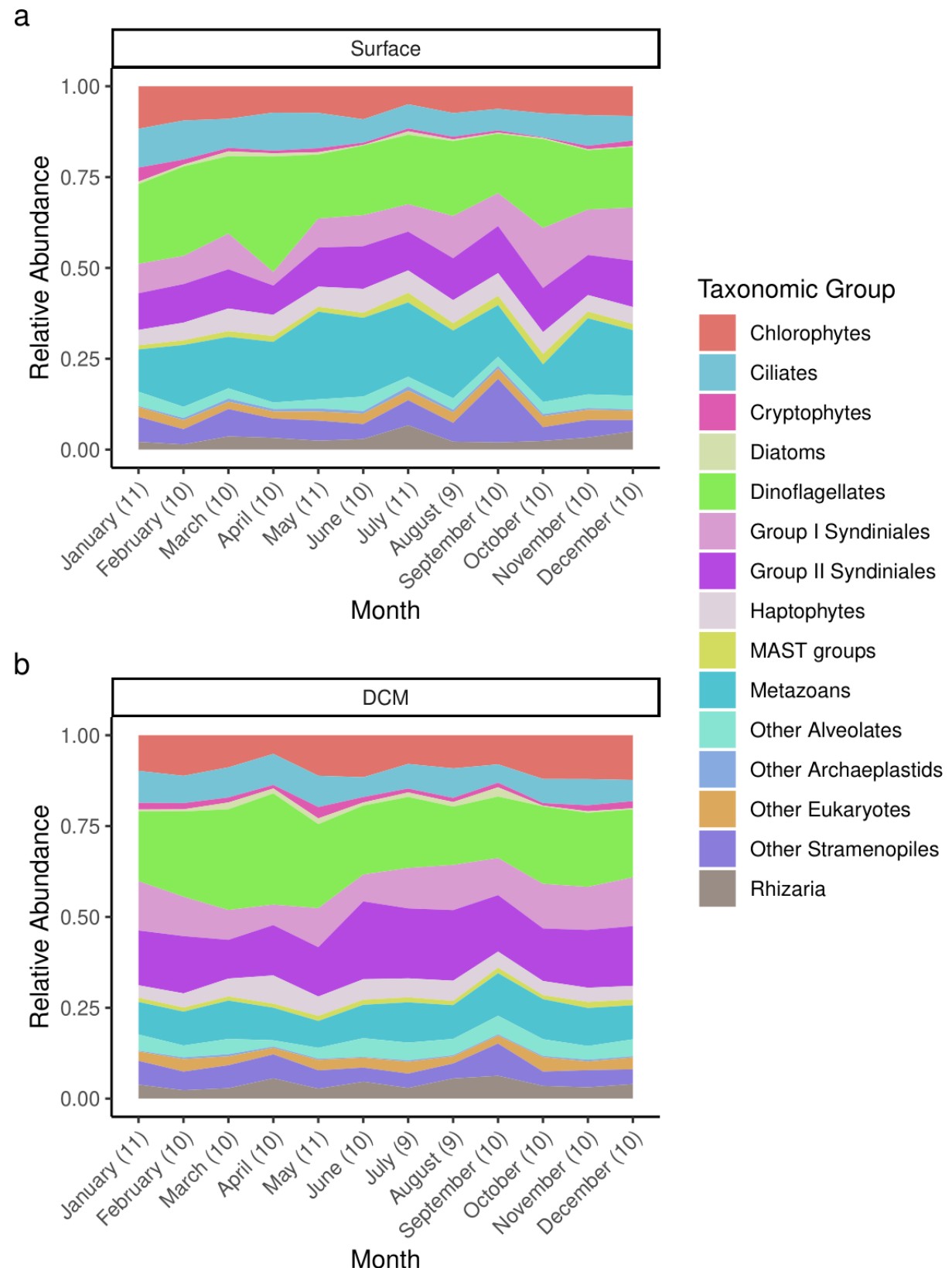

**FIG 1** Major taxonomic groups comprising the protistan communities observed each month at the SPOT station from 2003 to 2018. The taxonomic groups were manually curated. Panels show the average, monthly community composition at the surface (a) and at the DCM (b). Numbers in parentheses on the x-axis refer to the number of samples that were averaged for each month. Some months were missed during the 14-year and 9-month time series (see Materials and Methods for more details).

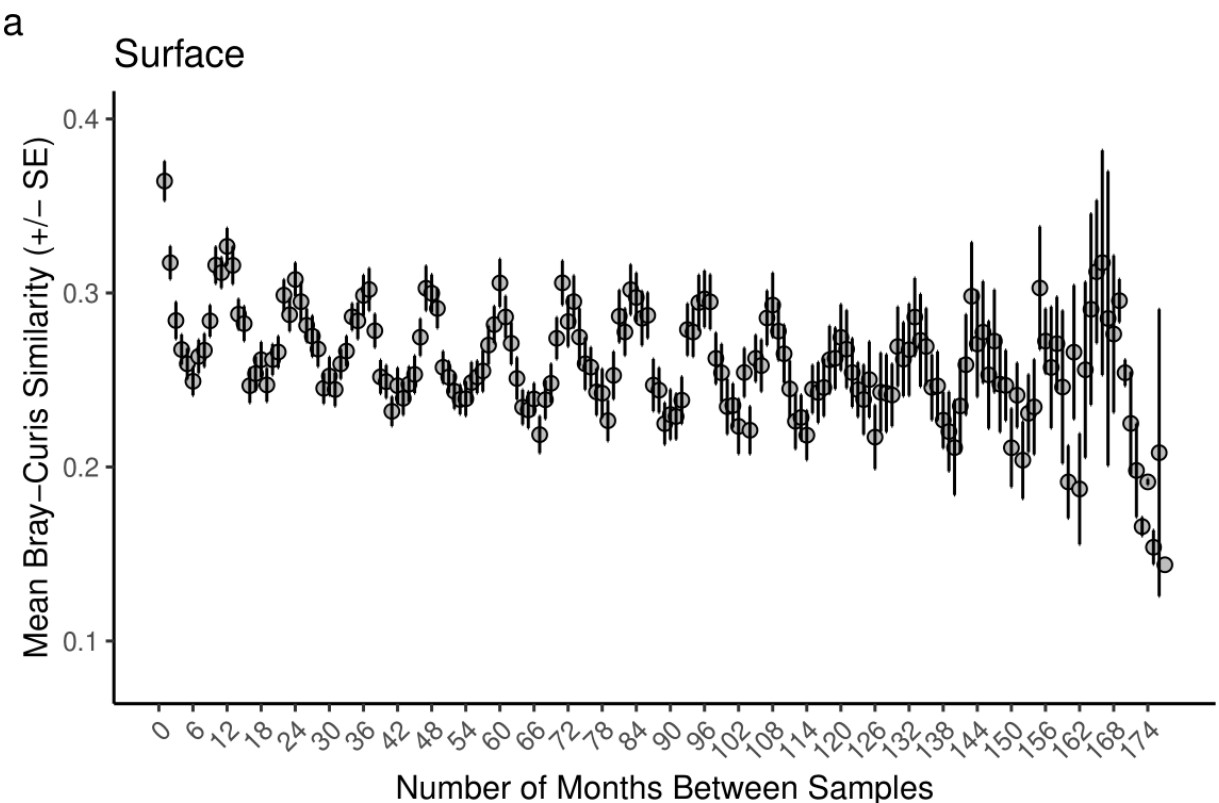

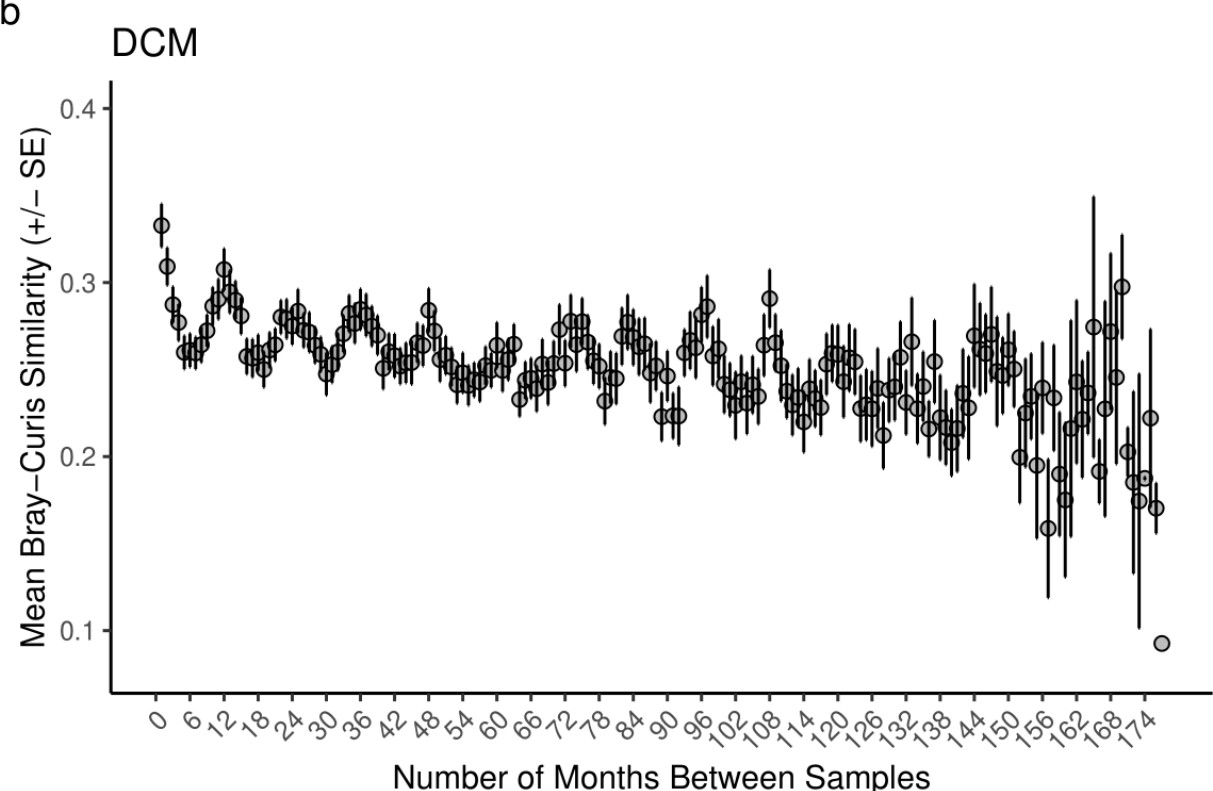

**FIG 2** Temporal patterns in surface (a) and DCM (b) protistan community similarity at the SPOT station. The *x*-axis values describe the number of months between the compared samples (1–177 months apart), while the *y*-axis values describe the average ± SE Bray-Curtis similarity for pairs of samples collected 1–177 months apart. The Bray-Curtis similarity values were calculated by subtracting the Bray-Curtis dissimilarity between every pair of samples from 1. Bray-Curtis

Fig 2 (Continued)

similarity values that compared samples across the same interval of time (1 month apart, 2 months apart, etc.) were then averaged and standard errors were calculated. Bray-Curtis similarity values closer to 1 indicate that the two samples compared were more similar to one another in terms of protistan community composition, while Bray-Curtis similarity values closer to 0 indicate that the two samples compared were less similar to one another.

community composition that were observed in the current study have been observed in previous SPOT studies (6, 10, 37, 38, 40) and have been roughly correlated with changes in physiochemical parameters such as seawater temperature and salinity (6).

While seasonal and interannual variability in protistan community composition was evident in the 14-year and 9-month SPOT time series, the environmental and commonly measured biological variables (Table S2) that were analyzed in this study only explained a small fraction of the variation in community composition (Fig. 3). Samples generally separated by season on the surface RDA plot (Fig. 3a); however, RDA explained <17% of the variability in community composition when taking all of the RDA axes into account (Table S3). There was substantial overlap in community composition for samples collected at the DCM in different seasons (Fig. 3b), and the RDA explained <15% of the variability in community composition when examining the percent of the variability explained by all RDA axes (Table S3). These findings indicate that the temporally variable environmental/biological parameters used in the ordinations could not explain most of the variation in protistan community composition observed throughout the time series (Fig. 3). This finding was expected, as a recent SPOT study by Yeh and Fuhrman (10) found that only 6–7% of the variability in protistan community composition could be explained by season at the surface and DCM. While the values reported by Yeh and Fuhrman (10) are not identical to those reported in the current study due to methodological differences in sequencing and bioinformatic analyses, both studies indicate that a large fraction (at least 83%) of the variability in protistan community composition cannot be explained by seasonal or long-term changes in the environment.

The minor influence that environmental/biological parameters had on protistan community structure at the SPOT station may be explained in part by the subtropical nature of the study site. Previous studies have revealed that seasonal and interannual shifts in the physiochemical environment at SPOT are subtle relative to environmental fluctuations that are evident at higher latitudes (37). Additionally, SPOT often resembles an oceanic environment and is therefore less influenced by upwelling relative to near-shore locations along the California coast which can result in greater seasonal extremes in productivity and planktonic biomass (64). Studies carried out in temperate or polar systems where the physical and chemical properties of waters undergo dynamic changes throughout the year have demonstrated that environmental parameters can account for a relatively large fraction of the variability in protistan community composition. For example, a 6-year time-series study conducted in Narragansett Bay (RI, USA) revealed that environmental parameters explained >80% of the variance in diatom community composition (65). Similarly, ≈55% of the variability in protistan community composition was explained by environmental parameters in a study carried out in the Beaufort Sea (AK, USA) (66). These observations indicate that the importance of environmental features in shaping protistan community structure is dependent on the degree of environmental variability over time.

## Network analyses recovered several well-characterized ecological associations and revealed numerous, putative, uncharacterized associations

Our network analyses were constructed using 14 years and 9 months of monthly, time-series ASV relative abundance data and environmental/biological data (Table S2) at the surface (i.e., 5 m) and the DCM (i.e., 5–58 m). These analyses yielded a total of 32,842 interactions at the surface and 33,190 interactions at the DCM. Only 2,644 interactions were observed in both network outputs (<9% of the interactions at each depth). In the surface network, there were 31,679 interactions between ASVs and other ASVs, and 1,098 interactions between ASVs and environmental/biological parameters. Similarly, in the

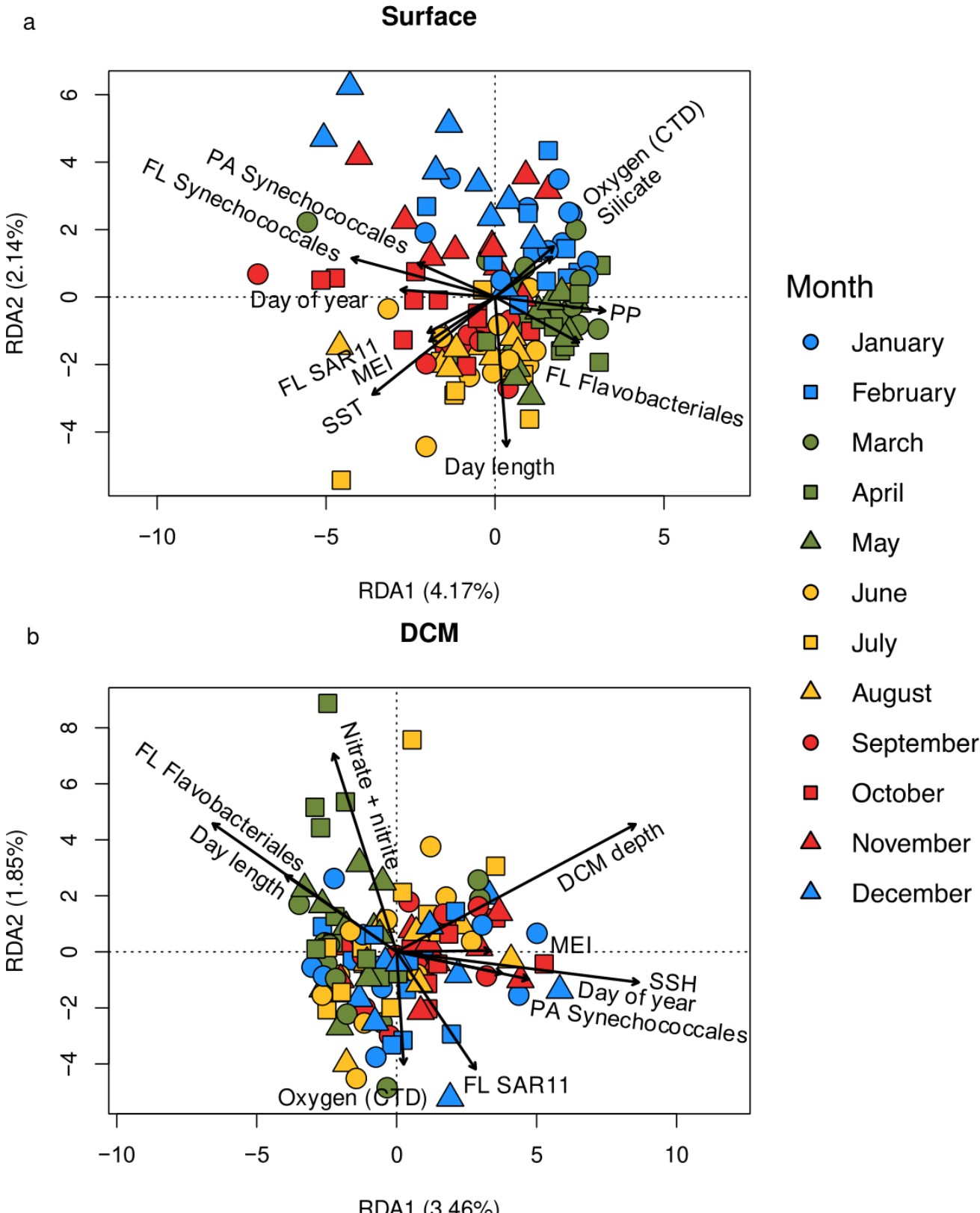

**FIG 3** Redundancy analysis (RDA) results for protistan communities from the surface (a) and DCM (b) at the SPOT station. The blue, green, yellow, and red shapes represent samples collected in winter, spring, summer, and autumn, respectively. The circles, squares, and triangles represent the different sample collection

Fig 3 (Continued)

months associated with each season. The RDAs were carried out using a forward selection procedure to determine the environmental and biological variables that explained a statistically significant ($P_{adj}$ < 0.05) portion of the variability in community composition. The environmental and biological variables that explained a significant fraction of the variability in community composition were projected as vectors onto the RDA plots (abbreviations: FL, free-living; MEI, multivariate ENSO index; PA, particle-associated; PP, satellite-derived primary production; SSH, sea surface height; SST, satellite-derived sea surface temperature). The RDA for surface samples explained a total of 16.3% of the variability in protistan community composition (including all RDA axes), while the RDA for DCM samples explained a total of 14.1% of the variability in protistan community composition (Table S3).

DCM network, there were 32,120 interactions between ASVs and other ASVs, and 1,015 interactions between ASVs and environmental/biological parameters. The PIDA-supported interactions comprised a small fraction of the ASV-ASV interactions, with 637 and 712 PIDA-supported interactions identified in the surface and DCM networks, respectively.

### Parasitic relationships were the most common interaction type

A large majority of the PIDA-supported associations identified in the network outputs were parasitic relationships (1,156 associations; ~81.5% of all PIDA-supported associations), emphasizing the potential importance of parasites in affecting protistan population abundances at the SPOT station. There were 539 putative parasitic relationships identified in the surface network (~84.6% of all PIDA-supported interactions at the surface) and 586 putative parasitic relationships identified in the DCM network (~82.3% of all PIDA-supported interactions at the DCM), most of which involved a Group I or Group II Syndiniales parasite ASV and a dinoflagellate or ciliate host ASV (Fig. 4; Fig. S4). There were also a few PIDA-supported parasitic relationships identified between *Cryothecomonas* (cercozoan) parasites and diatom host genera (Fig. 4; Fig. S4).

The SPOT networks identified more PIDA-supported parasitic interactions than were recorded in the PIDA database (Table 1). The higher number of PIDA-supported parasitic interactions identified in the surface and DCM networks relative to the number of

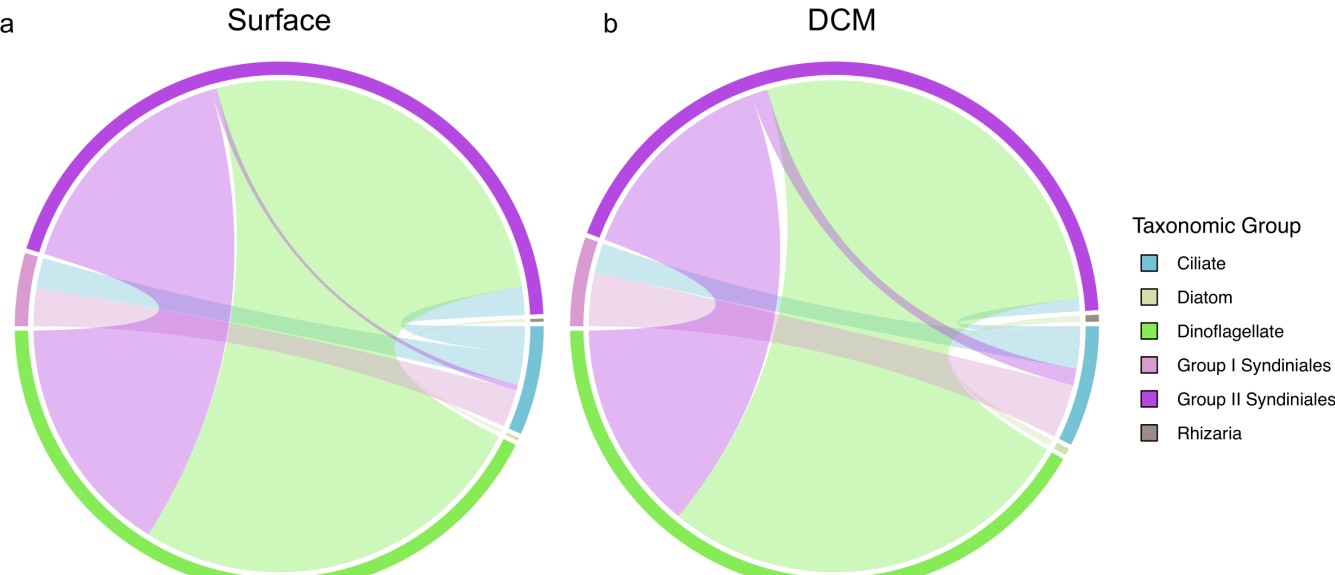

**FIG 4** Chord diagram subnetworks depicting the PIDA-supported parasitic relationships identified in the surface (a) and DCM (b) network outputs. Interactions involving Syndiniales parasites were considered PIDA-supported when the Syndiniales group and host genus in the Protist Interaction Database (PIDA) (13) matched a network prediction. All other parasitic interactions that were considered PIDA-supported matched a parasitic relationship documented in the PIDA database at the genus level. The colors delineate the major taxonomic affiliations of the ASVs involved in the parasitic relationships and the width of the lines connecting different taxonomic groups are proportional to the number of parasitic interactions that were recovered between taxonomic groups. The full taxonomic information for each ASV in these PIDA-supported parasitic subnetworks is documented in Table S4.

**TABLE 1** The number of parasitic, predator-prey, and "other symbiotic" relationships in PIDA, compared to the number of PIDA-supported parasitic, predator-prey, and "other symbiotic" interactions detected in the SPOT networks

| Parameter | Parasitic interactions | Predator-prey interactions | "Other symbiotic" interactions |
|---|---|---|---|
| Number of interactions in PIDA | 402 | 788 | 662 |
| Number of interactions in PIDA where both organisms were present in the network input | 65 | 110 | 22 |
| Number of PIDA-supported interactions in the SPOT surface network | 539 | 91 | 7 |
| Number of PIDA-supported interactions in the SPOT DCM network | 586 | 107 | 19 |

parasitic interactions recorded in the database can be explained by the differences in PIDA taxonomy and ASV taxonomy that were prevalent in our analyses. Syndiniales parasites are a diverse group of organisms, and the taxonomy of this group remains largely unresolved (67). Due to the difficulties associated with assigning taxonomic identifications to Syndiniales ASVs, we identified PIDA-supported Syndiniales relationships at the group level (as opposed to the genus or species level), which may have increased the number of false positive Syndiniales parasitic relationships that were recovered in our SPOT networks. Additionally, many Syndiniales parasites are thought to have multiple copies of the 18S rRNA gene (36), which can cause a single, Syndiniales-host interaction to be identified multiple times in the network output. These challenges associated with the identification of Syndiniales-host interactions will improve over time as more information on the taxonomy and ecology of these parasites is obtained.

The significant impacts that Syndiniales parasites and other parasites can have on protistan community composition have been highlighted in several studies (9, 15, 33, 35, 36, 67, 68). One recent study by Ollison et al. (35) used network analyses and light microscopy to identify putative associations between cercozoan *Cryothecomonas* parasites and diatom hosts in the Southern California Bight. The presence of these putative, parasitic relationships led Ollison et al. (35) to hypothesize that parasitism led to a decline in diatoms and a subsequent increase in dinoflagellates throughout a seasonal phytoplankton bloom period. Similarly, a study carried out by Berdjeb et al. (9) revealed that cercozoan parasites (*Cryothecomonas longipes* and *Protaspis grandis*) were found at high relative abundances at the end of a *Pseudo-nitzschia* bloom at the SPOT station, suggesting that these parasites may have played a role in the termination of the observed bloom.

### Dinophysis and gymnodinium engaged in kleptoplasty

Two of the PIDA-supported relationships that were detected in the SPOT network outputs were characterized as kleptochloroplastic interactions, whereby a non-photosynthetic organism acquires and retains a functional chloroplast from a photosynthetic organism (69). Specifically, in the surface network, an indirect, kleptochloroplastic relationship between a *Dinophysis* (dinoflagellate) ASV and a *Teleaulax acuta* (cryptophyte) ASV was observed (Fig. 5a; Fig. S5a), while in the DCM network, a direct kleptochloroplastic relationship between a *Gymnodinium* (dinoflagellate) ASV and a *Teleaulax acuta* (cryptophyte) ASV was observed (Fig. 5b; Fig. S5b). The *Dinophysis-Teleaulax* interaction is considered indirect because *Teleaulax* (a chloroplast-containing cryptophyte) is first consumed by *Mesodinium rubrum* (ciliate; previously known as *Myrionecta rubra*) and the functioning cryptophyte chloroplast is retained in *M. rubrum* during a primary kleptochloroplastic event (70–72). Subsequently, a secondary kleptochloroplastic event occurs when *M. rubrum* is consumed by *Dinophysis* and the functioning cryptophyte chloroplast is integrated into the *Dinophysis* cell (72, 73). Such kleptochloroplastic interactions benefit *Dinophysis*, *M. rubrum*, and *Gymnodinium* cells by allowing these organisms to utilize a mixotrophic nutritional strategy (phototrophy supplementing heterotrophy [73]).

The *Mesodinium* link in the indirect, kleptochloroplastic interaction was not detected in the SPOT network results because *Mesodinium* ASVs were not identified in the 18S-V4 time-series data (across all 29,123 ASVs). One potential explanation for the lack

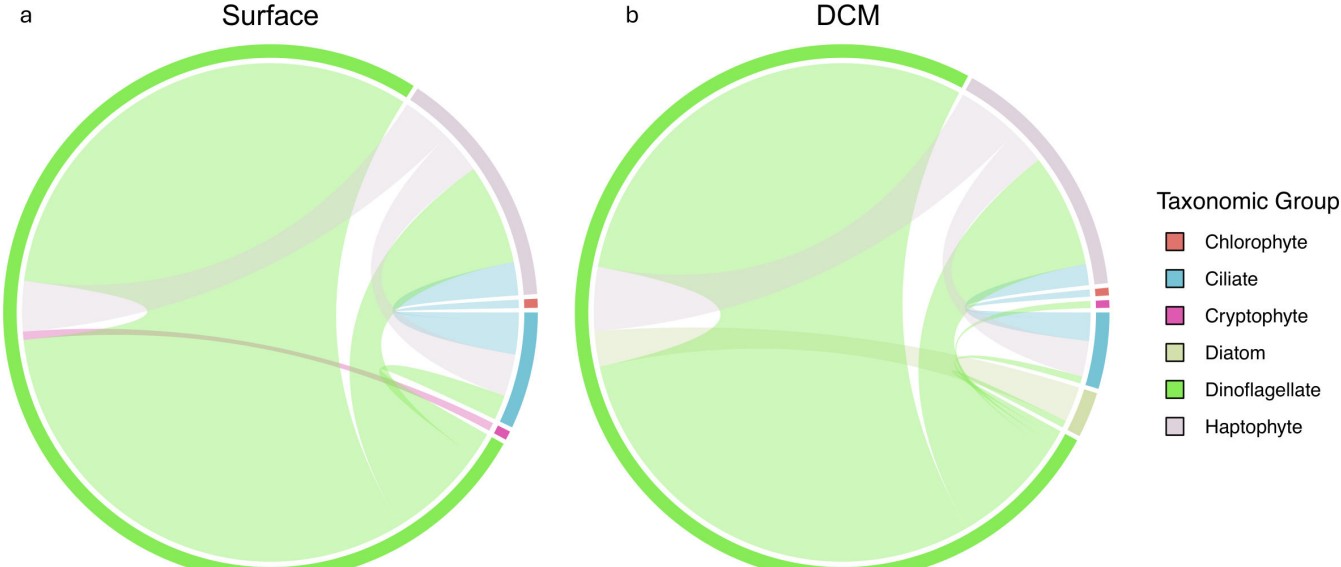

**FIG 5** Chord diagram subnetworks depicting the PIDA-supported predator-prey and kleptoplastic relationships identified in the surface (a) and DCM (b) network outputs. Network predictions that matched a predator-prey or kleptoplastic relationship recorded in the Protist Interaction Database (PIDA) (13) at the genus level were considered PIDA-supported. The colors delineate the major taxonomic affiliations of the ASVs involved in the predator-prey relationships and the width of the lines connecting different taxonomic groups are proportional to the number of predator-prey interactions that were recovered between taxonomic groups. The full taxonomic information for each ASV in these PIDA-supported subnetworks is documented in Table S5.

of *Mesodinium* ASVs in the data set is that the 18S-V4 primers that were used in this study did not amplify *Mesodinium* sequences due to deletions and structural variations in the V4 region of the 18S rRNA gene (74, 75). A previous study by Needham et al. (26) found evidence of a *Teleaulax-Mesodinium* relationship in surface waters near the SPOT station by using a higher-frequency sampling scheme, a different primer pair, and a different network analysis method than those used in the current study. Taken together, the results of Needham et al. (26) and the current study indicate that the "serial" kleptochloroplastic relationship between *Teleaulax*, *Mesodinium*, and *Dinophysis* is likely occurring *in situ* at SPOT, but that different sequencing methods and analyses may capture different components of this relationship.

### Acantharean and polycystine photosymbioses were detected

Several PIDA-supported photosymbiotic interactions were detected in the network outputs. Most of these photosymbioses occurred between acantharean hosts and a dinoflagellate or haptophyte symbiont (Fig. 6); however, a single photosymbiotic relationship was observed between a polycystine host and a dinoflagellate symbiont at the DCM (Fig. 6b). While acantharean and polycystine radiolarians are heterotrophic protists, some members of these groups harbor photosynthetic endosymbionts that confer mixotrophic ability on these organisms (73, 76, 77). Most acantharean ASVs involved in these putative photosymbiotic interactions in the current study were associated with more than one symbiont ASV, supporting the findings of studies that have reported a range of different symbiont types in a single acantharean cell (78, 79). The molecular underpinnings of acantharean photosymbioses are still being actively explored; however, evidence suggests that the algal endosymbionts do not benefit from the interaction and that these photosymbioses may therefore represent a form of parasitism (80).

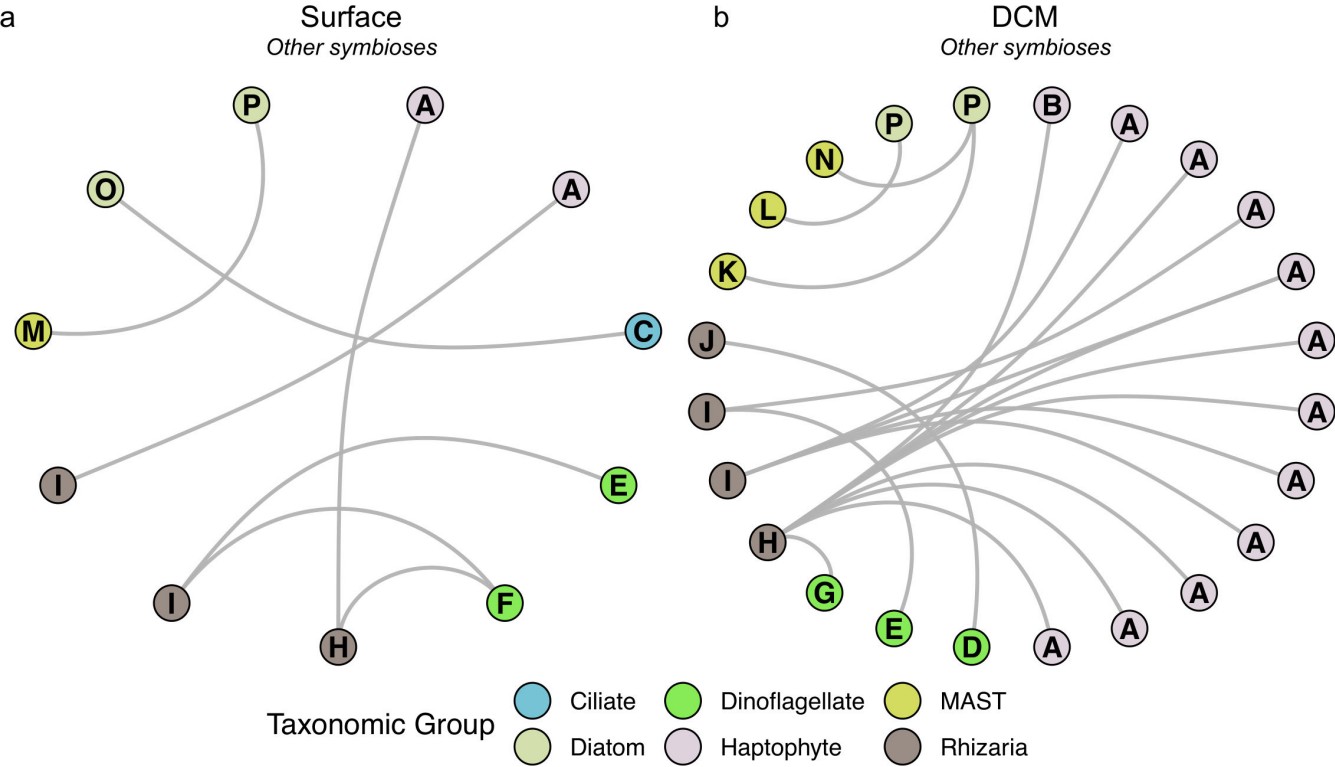

**FIG 6** Subnetworks depicting other PIDA-supported symbiotic interactions that were identified in the surface (a) and DCM (b) network outputs. Interactions involving acantharean and polycystine radiolarians were considered supported by the Protist Interaction Database (PIDA) (13) when an acantharean (Clade F) or polycystine host was associated with a known, symbiont genus. Interactions involving MAST cells were considered PIDA-supported when the MAST clade and the genus of the symbiont matched an association documented in the PIDA database. All other depicted network predictions matched a relationship recorded in the PIDA database at the genus level. Circles represent individual ASVs that were involved in network-predicted associations and the circle colors delineate the major taxonomic affiliations of the ASVs. Letters denote the genus-level or lowest-level taxonomic identification for all ASVs (A = *Chrysochromulina*, B = *Phaeocystis,* C = *Eutintinnus,* D = *Gymnodinium,* E = *Heterocapsa*, F = *Pelagodinium*, G = *Scrippsiella*, H = *Amphibelone*, I = Acantharea [Clade F], J = Polycystine [Nassellaria], K = MAST-3E, L = MAST-3F, M = MAST-3I, N = MAST-3L, O = *Chaetoceros*, P = *Leptocylindrus*). The full taxonomic information for each ASV in these PIDA-supported symbiotic subnetworks is documented in Table S6.

### Diatoms were involved in mutualistic epibioses

A symbiotic interaction between a *Eutintinnus* (tintinnid ciliate) ASV and a *Chaetoceros* (diatom) ASV was identified in the PIDA-supported SPOT surface network (Fig. 6a) and is thought to be mutualistic in nature (81, 82). This mutualistic relationship is also considered an epibiosis, in that the diatom attaches to and lives on the ciliate lorica (81). Hypotheses regarding the benefits of the *Eutintinnus-Chaetoceros* relationship have included strengthening of the ciliate lorica as a consequence of diatom attachment, and increased sinking speed of the cells perhaps aiding in the avoidance of predation (81). Additionally, Jonsson et al. (83) demonstrated that the attachment of diatoms to lorica could increase flow rates near the lorica, potentially increasing the feeding rates of the ciliates. The epibiosis may also increase *Chaetoceros* buoyancy and mobility, ultimately increasing the amount of light and nutrients available to these cells (81, 82).

Putative mutualistic epibioses between MAST-3 ASVs and *Leptocylindrus* (diatom) ASVs were also observed in the network outputs (Fig. 6). It has been suggested that the MAST-3 cells benefit from this MAST-3-*Leptocylindrus* relationship by obtaining shelter inside the diatom frustules, while the *Leptocylindrus* cells benefit by gaining motility through the motile MAST-3 cells (84, 85). A study by Gómez et al. (85) also suggested that there may be metabolic exchanges occurring between MAST-3 and *Leptocylindrus* cells involved in this epibiosis; however, this speculation awaits experimental confirmation.

## Approximately 2% of the network predictions were supported by the PIDA database

The PIDA-supported interactions comprised only 1.9% and 2.1% of the putative interactions identified in networks generated using surface and DCM samples, respectively. The relatively low number of network-predicted interactions that were found in the PIDA database may be partially due to the presence of false positive interactions in the network outputs. A previous study that used mock time-series data to carry out network analyses revealed that the precision (i.e., the number of real interactions in a network output divided by the total number of network predictions) of glasso time-series networks ranged from 11% to 29% when the abundance patterns of half of the organisms in the network exhibited seasonality (57). Similarly, a study that evaluated the precision of multiple correlation-based network analysis approaches found that network precision was <20% regardless of what network algorithm was used (86).

When both organisms involved in a PIDA-supported interaction were present in the SPOT network inputs, these interactions were typically captured in the SPOT network results (Table 1). A comparison of the number of interactions in PIDA where both organisms were present in the network input with the number of PIDA-supported interactions identified in the network results demonstrated that our analyses effectively captured many interactions occurring in nature, indicating relatively high network recall. Additionally, even with a conservative network precision estimate (e.g., 10%), we would expect thousands of the predictions in our SPOT networks to be true positive relationships that result from real, protist-protist interactions. Therefore, the relatively low percentage of PIDA-supported interactions in our SPOT networks may indicate that more work is needed to confirm and characterize many of the network predictions.

There are also hundreds of interactions in PIDA between protists and bacteria and protists and archaea that were not recovered in our SPOT networks, as our SPOT networks were constructed using only eukaryotic ASV relative abundances. Interactions between protists and other microorganisms can impact protistan community structure and function in several ways. For example, previous SPOT network analysis efforts have identified several putative predator-prey interactions between protistan grazers and prokaryotic taxa (31, 32). The abundances of key prokaryotic prey species could impact the relative abundances of protistan grazers, making these interactions important in shaping protistan community composition. Similarly, a previous study conducted at the SPOT station revealed that copepod grazing on protists can elicit significant top-down control on protistan populations, thus altering the composition of the observed community (87). Identifying and characterizing the interactions that may be occurring between protists and other microorganisms will require additional research efforts. In recent years, fluorescence *in situ* hybridization techniques (67, 88), imaging methodologies (11, 78, 89), and single-cell sorting and sequencing technologies (90, 91) have been used to identify and study putative microbial interactions. These approaches can be applied to confirm hypothesized associations recovered from network analyses, thus expanding the breadth of protist-protist, protist-bacteria, protist-archaea, and protist-metazoa interactions recorded in databases such as PIDA (13).

## Network predictions that were not present in the PIDA database may represent novel protist-protist interactions

Many of the network predictions that did not match an interaction recorded in the PIDA database were strongly correlated (SCC ≥ 0.64; Fig. 7; Fig. S6; Table S7) and may represent novel interactions that warrant further study. The co-occurrence patterns that were observed among these strongly correlated ASVs generally occurred across the entire study period (Fig. S6). While identifying the exact nature of these unresolved interactions is presently challenging, information on the physiology and ecology of these co-occurring organisms can be used to form hypotheses about the type of interaction that may be occurring between two correlated ASVs. For example, some of the strongly

Surface

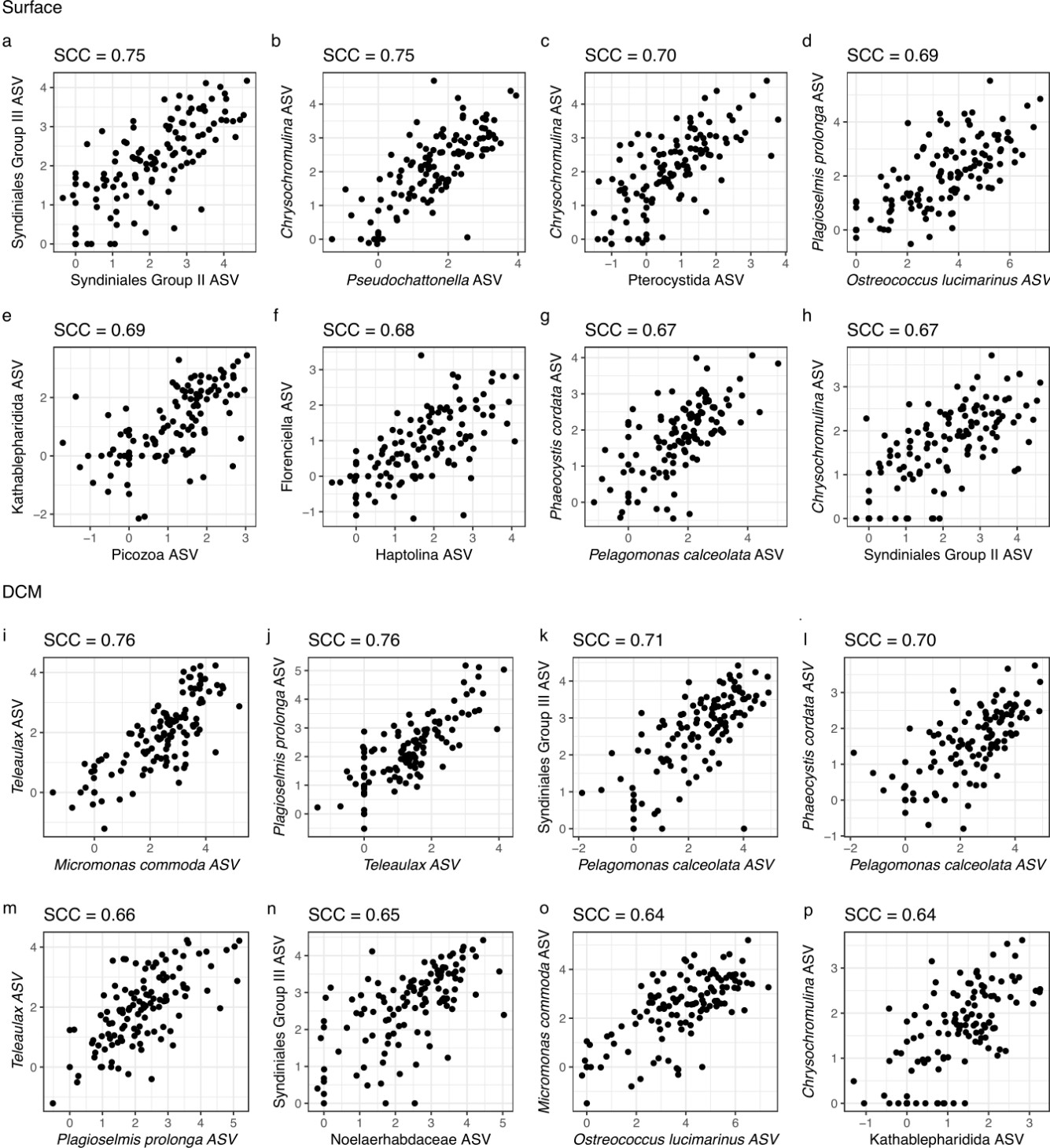

**FIG 7** Some of the ASV-ASV relationships that may represent novel interactions. The ASV abundance data were CLR-transformed, and the Spearman correlation coefficient (SCC) was calculated for pairs of ASVs that were involved in a network-predicted interaction. Some of the strongest network-predicted interactions at the surface (a–h) and DCM (i–p) that were not found in the PIDA database were visualized. The x- and y-axes are the CLR-transformed abundance of an individual ASV. Axes are labeled with the lowest level of taxonomy that was assigned to each ASV. The full taxonomic information for each ASV involved in these putative novel interactions is documented in Table S7.

correlated interactions that were recovered in the SPOT networks may be putative parasitic interactions involving Group III Syndiniales ASVs (Fig. 7; Fig. S6). The potential

for parasitism in Group III Syndiniales remains unknown (67); however, parasitism has been observed in Groups I and II Syndiniales, and recent studies have identified strong relationships between Group III Syndiniales and other protists (33, 92; Fig. 4; Table S4). There was also a strong, potentially novel interaction between a centrohelid (rhizarian) ASV (Pterocystida; Fig. 7c) and a *Chrysochromulina* (haptophyte) ASV at the surface that could represent a form of photosymbiosis similar to that observed between other rhizarian hosts and their photosymbiotic counterparts (e.g., acantharean hosts and haptophyte photosymbionts; Fig. 6). Some of the unresolved network predictions may also be related to predator-prey interactions. For example, an interaction detected at the DCM between a Kathablepharidida ASV and a *Chrysochromulina* ASV may result from Kathatablepharidida cells grazing on *Chrysochromulina*, as certain Katablepharid species have been shown to graze on *Chrysochromulina* (93). The network outputs also contained ASVs that were strongly correlated as a result of intraspecies diversity. Specifically, there were two strong (SCC = 0.76; SCC = 0.66) interactions that were observed between *Plagioselmis prolonga* ASVs and *Teleaulax* ASVs at the DCM that may be related to a dimorphic sexual life cycle in *Teleaulax* (Fig. 7j and m). A study carried out by Altenburger et al. (94) revealed that *P. prolonga* is not a unique cryptophyte species and is instead the co-occurring haploid stage of the *Teleaulax amphioxeia* dimorphic life cycle. The presence of strong *P. prolonga-Teleaulax* interactions in the DCM network output (Fig. 7j and m) demonstrates that network analysis tools have the potential to capture the unique forms of intraspecies diversity that may be prevalent in marine microbial communities.

Finally, the methods used in our SPOT network analyses were designed to decrease the influence that temporal co-occurrence patterns have on network inference (57). However, it is still possible that some of the unresolved, strong interactions captured in our networks represent the co-occurrence of two ASVs as opposed to a direct interaction. For example, the interaction detected between the picoeukaryotes *Micromonas commoda* and *Ostreococcus lucimarinus* (Fig. 7o) could result from temporal co-occurrence as opposed to a direct interaction. Previous studies have demonstrated that *Micromonas* and *Ostreococcus*, co-occur at SPOT and tend to peak in abundance in January, July, and October (38). Whether these picoeukaryote co-occurrence patterns result from shared environmental preferences, an indirect, biotic interaction, or a direct, biotic interaction will require further analyses of the ecology and physiology of these organisms at SPOT.

## Conclusion

The present study identified some of the important ecosystem properties and protist-protist interactions that played a role in shaping protistan community composition at the SPOT station over a 14-year and 9-month long period. Specifically, we found that the environmental/biological data explained a relatively small fraction of the total variability in protistan community composition. Our network analyses revealed numerous established (i.e., PIDA-supported) ecological interactions occurring between protists at the SPOT station and identified many putative ecological interactions that could be taking place but have not been established or characterized at this time. The PIDA-supported interactions included parasitic, kleptochloroplastic, predator-prey, photosymbiotic, and mutualistic interactions, emphasizing the diversity of interactions that occur between protists in the natural environment. These diverse interaction types contribute to the growth or demise of individual protist populations which, taken together, have a profound impact on the structure and composition of the entire protistan community.

## ACKNOWLEDGMENTS

The authors thank all individuals who contributed to the collection and analysis of SPOT data. The authors would especially like to thank Troy Gunderson, R/V Yellowfin captains and crew, Diane Kim, and Victoria Campbell for their support with the SPOT monthly sampling.

This work was supported by the National Science Foundation (grant OCE-1737409) to D.A.C. and J.A.F. and the Simons Foundation Collaboration on Computational Biogeochemical Modeling of Marine Ecosystems (CBIOMES) (grant 549943) to J.A.F. The funders of this work had no role in the study design, data collection, data analysis, or the decision to submit the work for publication.

## AUTHOR AFFILIATIONS

[1]Department of Biological Sciences, University of Southern California, Los Angeles, California, USA

[2]Department of Ecology, Evolution and Marine Biology, University of California Santa Barbara, Santa Barbara, California, USA

[3]Department of Marine Estuarine Environmental Science, Horn Point Laboratory, University of Maryland Center for Environmental Science, Cambridge, Maryland, USA

[4]Department of Ecology and Evolution, Stony Brook University, Stony Brook, New York, USA

[5]Institute for Advanced Computational Science, Stony Brook University, Stony Brook, New York, USA

[6]Department of Oceanography, Texas A&M University, College Station, Texas, USA

[7]Department of Global Ecology, Carnegie Institution for Science, Stanford University, Stanford, California, USA

## AUTHOR ORCIDs

Samantha J. Gleich http://orcid.org/0000-0002-4613-7171
J. L. Weissman http://orcid.org/0000-0002-4237-4807
Jed A. Fuhrman http://orcid.org/0000-0002-2361-1985

## FUNDING

| Funder | Grant(s) | Author(s) |
| --- | --- | --- |
| National Science Foundation (NSF) | OCE-1737409 | Jed A. Fuhrman |
| | | David A. Caron |
| Simons Foundation (SF) | 549943 | Jed A. Fuhrman |

## AUTHOR CONTRIBUTIONS

Samantha J. Gleich, Conceptualization, Data curation, Formal analysis, Methodology, Visualization, Writing – original draft, Writing – review and editing | Lisa Y. Mesrop, Formal analysis, Methodology, Writing – review and editing | Jacob A. Cram, Formal analysis, Writing – review and editing | J. L. Weissman, Formal analysis, Writing – review and editing | Sarah K. Hu, Formal analysis, Methodology, Writing – review and editing | Yi-Chun Yeh, Data curation, Methodology, Writing – review and editing | Jed A. Fuhrman, Funding acquisition, Writing – review and editing | David A. Caron, Conceptualization, Funding acquisition, Investigation, Supervision, Writing – review and editing

## DATA AVAILABILITY

The raw sequence data associated with this project can be obtained on BCO-DMO under data set number 770110 (http://lod.bco-dmo.org/id/dataset/770110). Code for the bioinformatic analyses, statistical analyses, and visualizations associated with this paper can be accessed at https://github.com/sgleich/SPOT_Protist_Net.

## ADDITIONAL FILES

The following material is available online.

## Supplemental Material

**Supplemental information (mSystems01045-24-s0001.docx).** Supplemental figures and captions for supplemental tables.
**Supplemental tables (mSystems01045-24-s0002.xlsx).** Tables S1 to S7.

## Open Peer Review

**PEER REVIEW HISTORY (review-history.pdf).** An accounting of the reviewer comments and feedback.

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
