## [Reviewer comments · mSystems]

With a little help from my friends: Importance of protist-protist interactions in structuring marine protistan communities in the San Pedro Channel

Samantha Gleich, Lisa Mesrop, Jacob Cram, Jake Weissman, Sarah Hu, Yi-Chun Yeh, Jed Fuhrman, and DAVID CARON

Corresponding Author(s): Samantha Gleich, University of Southern California

Review Timeline:

Submission Date:	August 1, 2024
Editorial Decision:	August 29, 2024
Revision Received:	December 7, 2024
Accepted:	December 26, 2024

Editor: Jeffrey Blanchard

Reviewer(s): Disclosure of reviewer identity is with reference to reviewer comments included in decision letter(s). The following individuals involved in review of your submission have agreed to reveal their identity: Sean Anderson (Reviewer #2)

Transaction Report:

DOI: <https://doi.org/10.1128/msystems.01045-24>

Re: mSystems01045-24 (With a little help from my friends: Importance of protist-protist interactions in structuring marine protistan communities in the San Pedro Channel)

Dear Dr. Samantha Gleich:

Revision Guidelines

Sincerely,
Jeffrey Blanchard
Editor
mSystems

Reviewer #1 (Comments for the Author):

The paper "With a Little Help from My Friends: Importance of Protist-Protist Interactions in Structuring Marine Protistan Communities in the San Pedro Channel" makes a significant contribution to the study of marine ecosystems, particularly by focusing on microbial eukaryotes - a group that often receives less attention compared to prokaryotic microorganisms.

Using a comprehensive amplicon dataset spanning fourteen years and nine months of monthly sampling in the San Pedro Ocean Time-series (SPOT), the authors have thoroughly investigated the influence of environmental variables and protist-protist

interactions on community composition. This extensive dataset provides a unique opportunity to assess the role of environmental and biological factors and shows that these alone do not explain most of the variability observed in protist communities.

The authors' focus on protist-protist interactions as a potential key factor in structuring communities is well-founded and adds a valuable dimension to our understanding of these complex systems.

Although I find the study convincing and the data solid, I have some concerns about the interpretation of the ecological networks.

Comments:

Broader context of interactions: The study does not address interactions beyond those between protists, which may be due to the limited data. However, top-down regulation by grazers or bottom-up effects by bacteria and viruses likely play an important role in shaping the community. Including the role of protist-protist interactions in a broader context of these additional interactions would have improved the introduction and discussion sections.

Validation of interaction predictions: the results, particularly the attempt to validate interaction predictions, are intriguing. However, I have some concerns about this aspect of the study. Would it be possible to reverse the process used in the study and analyze how many of the interactions described in the PIDA (assuming only marine interactions) were identified in the dataset when both partners were present? This could give a better insight into the predictive power of the network. It is currently unclear whether the PIDA interactions match well with those identified in the study network when the taxa were identified, but I assume that the interactions documented in the PIDA are generally confirmed by the data. It would also be beneficial to mention how many marine interactions there are in the PIDA, along with their categories (e.g. parasitism, predation), and whether these proportions are reflected in the predicted network of interactions, which would make it even clearer which categories of interactions are underrepresented. As interpretation and confidence in the interaction predictions is a critical issue, a more thorough investigation and discussion of the comparison with PIDA interactions would be valuable.

Unexplored Strong Interactions: I would appreciate a more in-depth discussion of the strong interactions that have been identified in the networks but are not present in PIDA. For example, how often (are they always present or at least all years in the same time?) can you recover strong interactions that you assume are crucial? Some interactions, such as between *Ostreococcus* and *Micromonas*, might represent co-occurrence rather than a direct interaction, and if not, what could the possible interaction be? A more detailed interpretation of the strong interactions not included in PIDA would be appreciated, as these could provide insights into currently overlooked aspects of protist community composition and perhaps provide a starting point for experimental work on these important interactions.

Overview:

In this study, Gleich et al. analyzed DNA samples collected monthly from the San Pedro Ocean Time-series (SPOT) station over a 14 year and 9-month period. They focused on 18S V4 primers, capturing biodiversity of protists at this site. They used statistical approaches to characterize temporal shifts in community composition, environmental drivers, and network associations in surface waters and in the deep chlorophyll maximum. Protist associations inferred from network analysis were compared against a known interaction database, which revealed that most associations at SPOT were not recorded and/or remained poorly described. It is important to identify putative relationships between protists that can seed further discovery and experimentation. This area of research is also important because microbes and their functional activities drive food web processes, biogeochemical cycles, and ecosystem stability.

Overall, I enjoyed the manuscript and feel that it will be appropriate for mSystems; however, I do have suggestions (many of them minor) that I feel may improve the paper before being considered for publication. Below are general and more specific comments and suggestions with line references.

General comments:

I know there has been a lot of data analyses with amplicon data collected at SPOT. Has this 18S V4 data been presented elsewhere in other papers? I think it is ok to analyze the data in a new way, but I would be clear in the Introduction or Methods section on what years of the dataset have been published already and in what manner the data has been analyzed (e.g., diversity metrics, networks, etc.). If it has been published, I think it is still awesome that you are leveraging the full temporal set to build networks and explore associations. I would just be clear and transparent on what has been documented and what this paper is adding to the story at SPOT.

You focused on protist interactions but also show some relative abundance data on metazoans. Did you filter out metazoan ASVs prior to the network analysis? I did not recall seeing this in the methods. I would likely exclude metazoans from the dataset because the core focus is on protists, and you already took measures to filter out larger organisms in the sample collection step. Either way, I would clarify the inclusion of metazoans in the paper and why or why not they were not considered in the networks.

Specific comments:

Line 31: What do you mean by biological variables? Is this referring to cell counts via microscopy? I may be more specific here. I am also not sure how the protist-protist interactions are influencing composition and if you show this. You show the impacts of metadata on composition but unclear how the interactions tie in directly with composition – these seem separate to me.

Line 40: Suggest adding "...that network analysis from amplicon datasets have the potential to capture true interactions". This statement is also a bit misleading, as only ~2% of your network connections were found in PIDA. Maybe rephrase to say that thousands of connections were found in PIDA but only 2% of the total interactions at SPOT were verified, and this highlights the need to better characterize protist interactions in natural marine environments.

Lines 90-92: This is true. However, such interactions can be inferred based on the organisms that are involved and the direction of that association (negative vs. positive).

Line 93: Might want to expand and say something like "...a list of known interactions that have been revealed and confirmed in culture or other controlled experiments".

Lines 96-98: I may go more active. "Here, we collected microbial eukaryotic amplicon sequence data...". For the lay person, you may want to specify at first mention that protists represent microbial eukaryotes. Then, stick to protists for the remainder of the paper, when possible, for consistency. I would also say microbial eukaryotes and not eukaryotes, as the story focuses on protist-protist relationships.

Lines 25-26: Can you provide an estimate on the number of missing months?

Line 140: I would mention that the relative abundance of specific prokaryotic taxa was determined via prior 16S metabarcoding datasets from the sample samples.

Line 176: QIIME 2 or qiime2?

Line 205: How did you standardize?

Lines 231-232: Why did you use 20% as the cutoff for the networks? Was this just an arbitrary cutoff? I am not saying there is anything wrong, but you may want to add a bit more as justification.

Line 235: Did you use all abiotic and biotic factors? Were there factors that were not significant in the RDA analysis that may be appropriate to exclude?

Line 242: I know you have done great work on NetGAM and validated this approach in a prior publication. I am just curious if you tried out other networks, like Spiec-Easi? Did your network approach address indirect edges or help to avoid large and dense networks? Does it get at negative vs. positive edges? I would just give a few more details on the approach and why it is the best (or a good option) in your case and with the dataset you have. Maybe discuss a sentence or so that summarizes the prior work you did on NetGAM?

Line 282: What does the nature of mean? Is this referring to characterizing ecological mechanisms behind some of the interactions?

Lines 292-294: I would be up front on the studies and years that utilized this dataset.

Line 295: Is this referring to the species level?

Line 303: Suggest changing to “across different time intervals”.

Line 308: Seasonal dynamics are well known at SPOT. This confirms that seasonal patterns are important over decadal scales.

Line 333: Suggest removing “that was observed”.

Line 334: Suggest removing “this”.

Line 335: Was this the sum of the first two RDA axes?

Line 342: How many years were considered in this paper? Yeh and Fuhrman used 16S primers that capture some protists, right? I would highlight somewhere that your 18S primers will pick up some important groups, like Syndiniales, MAST, ciliates, and radiolarians, that may be missed by universal 16S primers.

Lines 358-360: This thought seems to end abruptly. I would add a closing sentence or two and bring it back to your findings or just mention that sampling location may dictate the contribution of environmental factors in driving microbial communities.

Line 363: And these networks incorporated all monthly data over the 14 years at both the surface and DCM? I would highlight that here again. Did your network approach test for significance of the associations or give some sort of confidence value to any given ASV-ASV connection?

Line 372: It might be nice to give the percent that this represented in your networks along with the actual number.

Line 374: May be nice to give a percent of the PIDA supported associations that involved parasites.

Lines 376-377: I would specify parasite ASV and ciliate host ASV.

Lines 377-378: These were PIDA supported?

Line 387: High relative abundances?

Line 414: This is a personal preference, but I don't think you need to say “study” when referring to these papers. I would just say Needham et al.

Line 417: Add “methods and analysis...”.

Line 426: Did you measure network degree or other network stats?

Line 429: Remove “the” in “...the algal endosymbionts”.

Lines 431: The headings in this section could be more direct. Something like, “Diatoms involved in mutualistic epibioses.”

Line 444: Remove “the” in “...the MAST-3 cells...”.

Lines 447-449: These are cool findings, but I wonder if they could also be heterotrophic relationships at times. How can you be certain that these protist grazers are not consuming these prey items? I would include another sentence to clarify that this may also reflect grazing.

Lines 451-453: This sentence seems contradictory. You say many of them were identified but then mention that it was only ~2%. I think it is ok to say that only 2% of the associations were revealed in PIDA at the level that you could resolve in your amplicon data; however, your approach allowed for identification of putative association that may reflect actual interactions and spur additional studies to verify them. The point is that most interactions in situ are unresolved, and so, more studies like yours will be important to guide future efforts.

Line 455: Can you deal with false positives in NetGAM? I know that programs like Spiec-Easi aim to mitigate false positives and indirect relationships.

Line 463: What do you mean by “real”?

Line 485: What is meant by strong? I would be quantitative if possible.

Line 492: I think you are missing a section on broader impacts of the work and what your results would mean for ecosystems and carbon cycles.

Line 507: This last sentence should be expanded in the Discussion in a section where you mention implications and next steps.

Figure 1: I think it is fine that your taxonomic groupings do not follow a specific level in PR2, but I would clarify in the legend or methods that you grouped them manually into these categories. For instance, all diatoms and MAST were grouped but you separated out Syndiniales into the two major groups. Do the “other” groupings reflect a certain taxonomic level and relative abundance cutoff? Why keep metazoans in the analysis, if you were trying to filter them out in the first place? I would remove them, as the paper is focused on protist-protist associations. Shouldn't the number of samples in each month be 14? One for each year of the dataset? Oh, but some samples were removed. I would clarify in the legend.

Figure 3: I like this figure, but it is a bit small and hard to read. Can you increase the figure size and make the data points and vectors larger? The shapes and colors distinguish month and season, correct? It took me a minute to figure that out. What factors were significant to the RDA? Did this vary between surface and DCM? I would clarify in the legend and mention in the paper somewhere what factors (if any) were found to not be significant in the stepwise selection.

Figure 4: I like the idea, but this figure is hard to interpret. The nodes are way too small, and so, we cannot see the colors and taxonomic groups that they represent. Is there another way to summarize these connections? Or maybe move this to supplemental and focus on the top 10-20 connections? You could also include a venn diagram to show the overlap between SPOT and PIDA parasitic associations. The connections in panels a and c are easier to interpret if you increased the figure size. However, panels b and d are hard to interpret with all the gray lines present. Maybe try a chord type plot and color based on the type of connection. This may help to avoid all the single connecting lines.

Figure 5: This plot is larger and easier to read, though it does look a bit warped to me. I would suggest trying out chord plots that may improve the figure and make it easier to read. I appreciate the colors and letters to distinguish node, but it takes a long time staring at the figure to interpret what is happening. I would try a chord plot and see if that makes it any clearer.

Figure 7: All spearman values in this figure were significant, right? I would give the ASV number as well in case some of the same ASVs were found twice or they were unique but had the same taxonomic assignment. Maybe that situation never occurred?

mSystems01045-24

With a little help from my friends: Importance of protist-protist interactions in structuring marine protistan communities in the San Pedro Channel

Reviewer Responses

We thank the editor and reviewers for taking the time to read our paper and for providing great feedback. We have revised the manuscript based on the reviewer comments and we feel that these revisions have improved the presentation and discussion of our results.

Reviewer #1

The paper "With a Little Help from My Friends: Importance of Protist-Protist Interactions in Structuring Marine Protistan Communities in the San Pedro Channel" makes a significant contribution to the study of marine ecosystems, particularly by focusing on microbial eukaryotes - a group that often receives less attention compared to prokaryotic microorganisms.

Using a comprehensive amplicon dataset spanning fourteen years and nine months of monthly sampling in the San Pedro Ocean Time-series (SPOT), the authors have thoroughly investigated the influence of environmental variables and protist-protist interactions on community composition. This extensive dataset provides a unique opportunity to assess the role of environmental and biological factors and shows that these alone do not explain most of the variability observed in protist communities.

The authors' focus on protist-protist interactions as a potential key factor in structuring communities is well-founded and adds a valuable dimension to our understanding of these complex systems.

Although I find the study convincing and the data solid, I have some concerns about the interpretation of the ecological networks.

We thank the reviewer for their kind words and their insight on how to improve the interpretation of the ecological networks. We have incorporated the reviewer's ideas into our revised manuscript, and we feel that these changes have greatly improved the paper.

Comments:

Broader context of interactions: The study does not address interactions beyond those between protists, which may be due to the limited data. However, top-down regulation by grazers or bottom-up effects by bacteria and viruses likely play an important role in shaping the community. Including the role of protist-protist interactions in a broader context of these additional interactions would have improved the introduction and discussion sections.

Our study was focused on understanding the interactions among protists, and therefore constrained to those species (which was already a rather formidable dataset), but we appreciate the reviewer mentioning the importance of interactions that are occurring between protists and

other groups of organisms (e.g., bacteria-protist interactions, protist-metazoan interactions, etc.), and we agree that these interactions are likely important in shaping marine protistan communities. We have incorporated additional information into our Introduction and Results and Discussion sections (lines 87-88, lines 718-735) to emphasize the important role that these other interaction types play in structuring marine protistan communities. We believe that this extra information helps provide context for our protist-protist interaction network results.

Validation of interaction predictions: the results, particularly the attempt to validate interaction predictions, are intriguing. However, I have some concerns about this aspect of the study. Would it be possible to reverse the process used in the study and analyze how many of the interactions described in the PIDA (assuming only marine interactions) were identified in the dataset when both partners were present? This could give a better insight into the predictive power of the network. It is currently unclear whether the PIDA interactions match well with those identified in the study network when the taxa were identified, but I assume that the interactions documented in the PIDA are generally confirmed by the data. It would also be beneficial to mention how many marine interactions there are in the PIDA, along with their categories (e.g. parasitism, predation), and whether these proportions are reflected in the predicted network of interactions, which would make it even clearer which categories of interactions are underrepresented. As interpretation and confidence in the interaction predictions is a critical issue, a more thorough investigation and discussion of the comparison with PIDA interactions would be valuable.

We appreciate the reviewer pointing out that more information on the protist-protist interactions that currently exist in PIDA is needed. We have added additional information to our Materials and Methods section (lines 356-374) to describe the categories (e.g., parasitism, predation, etc.) that currently exist in PIDA and the frequency of those interaction types in the database.

In addition to adding information on the interactions that are recorded in PIDA, we also added some information to our Results and Discussion section based on the reviewer's interesting PIDA comparison ideas. We added a table (Table 1) and additional information to the Results and Discussion section (lines 537-550; lines 689-696) to compare the number of interactions of a given category that currently exist in PIDA to the number of PIDA-supported interactions of that category that were recovered in the SPOT networks. We also added information about how frequently a PIDA-supported interaction type was recovered in the SPOT networks when both taxa involved in an interaction were present in the input dataset. We believe that these additional comparisons to PIDA are great additions to our network validation approach.

Unexplored Strong Interactions: I would appreciate a more in-depth discussion of the strong interactions that have been identified in the networks but are not present in PIDA. For example, how often (are they always present or at least all years in the same time?) can you recover strong interactions that you assume are crucial? Some interactions, such as between *Ostreococcus* and *Micromonas*, might represent co-occurrence rather than a direct interaction, and if not, what could the possible interaction be? A more detailed interpretation of the strong interactions not included in PIDA would be appreciated, as these could provide insights into currently overlooked aspects of protist community composition and perhaps provide a starting point for experimental work on these important interactions.

We agree that the unexplored, strong interactions are intriguing and are an important starting point for future research endeavors. We have added additional information to our Results and Discussion section on the potential nature of the unexplored, strong interactions that were highlighted in Figure 7. We also plotted the ASVs that were involved in these unexplored, strong interactions over time to visualize the relative abundance patterns of pairs of organisms that were potentially interacting throughout the time-series (Figure S6).

Reviewer #2

Overview:

In this study, Gleich et al. analyzed DNA samples collected monthly from the San Pedro Ocean Time-series (SPOT) station over a 14 year and 9-month period. They focused on 18S V4 primers, capturing biodiversity of protists at this site. They used statistical approaches to characterize temporal shifts in community composition, environmental drivers, and network associations in surface waters and in the deep chlorophyll maximum. Protist associations inferred from network analysis were compared against a known interaction database, which revealed that most associations at SPOT were not recorded and/or remained poorly described. It is important to identify putative relationships between protists that can seed further discovery and experimentation. This area of research is also important because microbes and their functional activities drive food web processes, biogeochemical cycles, and ecosystem stability. Overall, I enjoyed the manuscript and feel that it will be appropriate for mSystems; however, I do have suggestions (many of them minor) that I feel may improve the paper before being considered for publication. Below are general and more specific comments and suggestions with line references.

We thank the reviewer for their kind words and their valuable suggestions. We have incorporated the reviewer's ideas into our revised manuscript, and we feel that these changes and additions have improved the manuscript.

General comments:

I know there has been a lot of data analyses with amplicon data collected at SPOT. Has this 18S V4 data been presented elsewhere in other papers? I think it is ok to analyze the data in a new way, but I would be clear in the Introduction or Methods section on what years of the dataset have been published already and in what manner the data has been analyzed (e.g., diversity metrics, networks, etc.). If it has been published, I think it is still awesome that you are leveraging the full temporal set to build networks and explore associations. I would just be clear and transparent on what has been documented and what this paper is adding to the story at SPOT.

We thank the reviewer for pointing out that there have been previous publications that have characterized protistan community composition at SPOT. Many of these previous efforts at SPOT have used molecular techniques such as terminal restriction fragment length polymorphism, metatranscriptomics, and 515Y-926R amplicons to characterize protistan community composition. The 14 year and 9-month 18S-V4 amplicon dataset leveraged in this study differs from previous datasets that have been highlighted in other publications.

A small subset of the amplicon data that were used in our analyses were used in a publication by Hu et al. (2016) that focused on RNA:DNA ratios at SPOT over a 1-year period (2013-2014). The other ~13 years of data included in this analysis have not been used in previous SPOT publications. We have revised our Materials in Methods section to include information about the Hu et al. (2016) publication (line 166-170).

You focused on protist interactions but also show some relative abundance data on metazoans. Did you filter out metazoan ASVs prior to the network analysis? I did not recall seeing this in the methods. I would likely exclude metazoans from the dataset because the core focus is on protists, and you already took measures to filter out larger organisms in the sample collection step. Either way, I would clarify the inclusion of metazoans in the paper and why or why not they were not considered in the networks.

We thank the reviewer for pointing out that our inclusion of metazoan ASVs was unclear. We chose to include metazoan ASVs in our networks because we used a graphical lasso networking approach. The graphical lasso network analysis uses a conditional independence-based framework to avoid capturing indirect interactions that may occur between two ASVs. For example, if a copepod grazes on two different diatom species, the abundance patterns of these diatoms may track each other because they are both influenced by the abundance of the copepod. If the copepod is not included in the network analysis, these two diatoms may appear to be “interacting”. By including metazoan ASVs in our networks, we may avoid capturing interactions that result from protistan ASVs that are both associated with the same metazoan ASV.

Specific comments:

Line 31: What do you mean by biological variables? Is this referring to cell counts via microscopy? I may be more specific here. I am also not sure how the protist-protist interactions are influencing composition and if you show this. You show the impacts of metadata on composition but unclear how the interactions tie in directly with composition –these seem separate to me.

We thank the reviewer for pointing out that our description of biological variables was unclear here. The “biological” variables we leveraged were variables that can change due to changes in protistan physiology such as chlorophyll *a* fluorescence and satellite-derived estimates of primary production. Our biological variables also included the mean relative abundances of specific bacterial orders (Table S2). We were limited by word count in the abstract, and therefore could not explain in detail what measurements were included in our analysis of “biological variables” in this section. Our Materials and Methods section (lines 195-201) explains the differences between the environmental and biological variables in more detail and Table S2 describes which factors we considered “environmental” and which factors we considered “biological” in this study.

Line 40: Suggest adding “...that network analysis from amplicon datasets have the potential to capture true interactions”. This statement is also a bit misleading, as only ~2% of your network connections were found in PIDA. Maybe rephrase to say that thousands of connections were

found in PIDA but only 2% of the total interactions at SPOT were verified, and this highlights the need to better characterize protist interactions in natural marine environments.

We have revised this line as the reviewer suggested (line 39-42).

Lines 90-92: This is true. However, such interactions can be inferred based on the organisms that are involved and the direction of that association (negative vs. positive).

We agree with the reviewer that sometimes these interactions can sometimes be inferred, and we have revised this sentence to reflect this (line 124-126). However, we would like to highlight that it can be difficult to use the direction of an association to categorize an interaction type. For example, a parasite and host may increase in abundance together over time (i.e., positively correlated) until some threshold host density is reached. After that threshold is reached, the relationship between parasite and host may become negative (i.e., parasite abundances increase, while host abundances decrease). These directional changes can make it difficult to characterize an interaction type using the direction of a correlation.

Line 93: Might want to expand and say something like "...a list of known interactions that have been revealed and confirmed in culture or other controlled experiments".

We have revised this line as the reviewer suggested (line 127-128).

Lines 96-98: I may go more active. "Here, we collected microbial eukaryotic amplicon sequence data...". For the lay person, you may want to specify at first mention that protists represent microbial eukaryotes. Then, stick to protists for the remainder of the paper, when possible, for consistency. I would also say microbial eukaryotes and not eukaryotes, as the story focuses on protist-protist relationships.

We thank the reviewer for these suggestions. We have revised this line (now line 131) accordingly and we have clarified in the manuscript that we focused on microbial eukaryotes (protists) in our analyses.

Lines 25-26: Can you provide an estimate on the number of missing months?

We have added information on which samples are missing from our 14 year and 9-month time-series (lines 174-175).

Line 140: I would mention that the relative abundance of specific prokaryotic taxa was determined via prior 16S metabarcoding datasets from the sample samples.

We clarified that the relative abundances of the prokaryotic taxa were obtained from previous SPOT sampling efforts (line 219).

Line 176: QIIME 2 or qiime2?

We have revised this section (lines 245-254).

Line 205: How did you standardize?

We have provided additional information on how the environmental and biological data were standardized prior to carrying out the PCA (lines 286-287).

Lines 231-232: Why did you use 20% as the cutoff for the networks? Was this just an arbitrary cutoff? I am not saying there is anything wrong, but you may want to add a bit more as justification.

When selecting a cutoff threshold, we aimed to include as many ASVs as possible, ensuring a broad representation of taxa while keeping the networking procedure computationally feasible. We added additional information on this threshold cutoff to our Materials and Methods section (lines 324-326).

Line 235: Did you use all abiotic and biotic factors? Were there factors that were not significant in the RDA analysis that may be appropriate to exclude?

We used all biotic and abiotic factors in our network analyses. Although there were abiotic/biotic factors that were not significantly associated with changes in protistan community composition (factors that were not projected as vectors in Figure 3), it is still possible that these abiotic/biotic factors could influence the abundance patterns of an individual ASV. Therefore, we chose to include all factors as input in our networks.

Line 242: I know you have done great work on NetGAM and validated this approach in a prior publication. I am just curious if you tried out other networks, like Spiec-Easi? Did your network approach address indirect edges or help to avoid large and dense networks? Does it get at negative vs. positive edges? I would just give a few more details on the approach and why it is the best (or a good option) in your case and with the dataset you have. Maybe discuss a sentence or so that summarizes the prior work you did on NetGAM?

We thank the reviewer for asking follow-up questions on our networking methods. We have added additional information on our network analyses to the Materials and Methods section (line 344-349).

In short, the NetGAM approach is not a network analysis by itself. NetGAM is a statistical data transformation that decreases the influence that time-series features (e.g., seasonality, long-term changes in abundance) have on microbial abundances. After applying a CLR transformation (to account for the relative nature of the ASV data) and the NetGAM transformation (to account for the temporal dependence of the ASV data), we used a graphical lasso network inference approach. This graphical lasso network inference approach is what the Spiec-Easi package is based on. Therefore, like Spiec-Easi, our network approach assumes that the true network is sparse, and it avoids capturing indirect associations between variables by using the concept of conditional independence.

Line 282: What does the nature of mean? Is this referring to characterizing ecological mechanisms behind some of the interactions?

Here we were referring to the type of interaction that is occurring between two organisms (e.g., parasitism, predation, etc.). We recognize that our language was vague, and we have clarified what we mean by this in our manuscript (lines 409-411).

Lines 292-294: I would be up front on the studies and years that utilized this dataset.

As mentioned above, most of the data presented in this manuscript (with the exception of 4 samples from 2013-2014) have not used in previous analyses. Many of the previous publications that have focused on protistan community composition at SPOT have used different methods to obtain relative abundance data. We have clarified this in our Materials and Methods section (lines 166-170) and in our Results and Discussion section (lines 424-425).

Line 295: Is this referring to the species level?

Not necessarily. The use of ASVs for characterizing microbial species richness has supplanted the use of OTUs (Operational Taxonomic Units). Neither approach guarantees that 1 unit = 1 species. However, ASVs are reproducible entities (once sequencing errors are eliminated), whereas OTUs are more dependent on the level of sequence similarity employed in the analysis. Conventional wisdom suggests that for many taxa (particularly many protists), two or more ASVs can have the same species-level identification. This means that ASVs can capture intraspecies diversity at times. While not ideal, it is presently the state-of-the-art when using sequence information to define microbial taxa.

Line 303: Suggest changing to “across different time intervals”.

This line has been changed accordingly (line 434).

Line 308: Seasonal dynamics are well known at SPOT. This confirms that seasonal patterns are important over decadal scales.

We thank the reviewer for pointing out that our data show seasonality being prevalent over decadal scales. We have added text to emphasize this in our Results and Discussion (line 440-441).

Line 333: Suggest removing “that was observed”.

We have revised this line accordingly (line 472).

Line 334: Suggest removing “this”.

We have revised this line accordingly (line 473).

Line 335: Was this the sum of the first two RDA axes?

We thank the reviewer for pointing out that our discussion of the RDA results is unclear. The percentages that were reported in the manuscript (~17% and ~15%; original manuscript lines 335-337) were referring to the sum of all of the RDA axes in multi-dimensional space (reported in Table S3). We have clarified this in our Results and Discussion section (lines 474-483).

Line 342: How many years were considered in this paper? Yeh and Fuhrman used 16S primers that capture some protists, right? I would highlight somewhere that your 18S primers will pick up some important groups, like Syndiniales, MAST, ciliates, and radiolarians, that may be missed by universal 16S primers.

Yeh and Fuhrman used universal 515Y-926R primers in their SPOT analyses, which have the potential to capture both prokaryotic and eukaryotic taxa including Syndiniales, MAST cells, ciliates, etc. While that is a useful approach, the resulting sequences are dominated by 16S sequences, sometimes leaving the 18S sequence datasets a bit 'sparse'. For that reason, we chose the more traditional 18S primers which eliminate much of the 16S dominance. In this sense, the 18S-V4 primers used in our analysis provide a somewhat different view of the eukaryotic taxa that are present at SPOT.

Lines 358-360: This thought seems to end abruptly. I would add a closing sentence or two and bring it back to your findings or just mention that sampling location may dictate the contribution of environmental factors in driving microbial communities.

We added a closing sentence to this paragraph (lines 508-509).

Line 363: And these networks incorporated all monthly data over the 14 years at both the surface and DCM? I would highlight that here again. Did your network approach test for significance of the associations or give some sort of confidence value to any given ASV-ASV connection?

Yes, the network analyses we carried out included all samples collected over a ~14-year period at both the surface and DCM. We have reiterated this when presenting the network results to ensure that the information is clear (lines 512-514).

The graphical lasso network approach we used for our analyses does not provide any confidence values to the connections that are identified in the networks. However, we agree that having metrics of correlation and significance may be useful and interesting to a reader. Therefore, we calculated the spearman correlation between every PIDA-supported ASV pair in our networks and added the correlation coefficient and p-value information in Tables S4-S6.

Line 372: It might be nice to give the percent that this represented in your networks along with the actual number.

We provided the number and percent of parasitic interactions in the PIDA-supported network outputs (line 519-531).

Line 374: May be nice to give a percent of the PIDA supported associations that involved parasites.

We added information on the percent of PIDA-supported interactions that were classified as parasitism to our Results and Discussion (line 524-533).

Lines 376-377: I would specify parasite ASV and ciliate host ASV.

We clarified this in our writing (line 533-534).

Lines 377-378: These were PIDA supported?

Yes, these interactions between *Cryothecomonas* parasites and diatoms hosts were PIDA-supported. We have added additional text to clarify this (line 535-536).

Line 387: High relative abundances?

We have revised this text for clarity (line 581).

Line 414: This is a personal preference, but I don't think you need to say "study" when referring to these papers. I would just say Needham et al.

We have removed the word "study" as the reviewer suggested to limit unnecessary wordiness in our writing.

Line 417: Add "methods and analysis...".

We have added this text to ensure clarity (line 618-619).

Line 426: Did you measure network degree or other network stats?

We did not calculate the degree or other network statistics. We were interested in focusing the discussion of our network analysis results on the PIDA comparisons to identify specific types of interactions, and to differentiate previously confirmed interactions from those that may represent novel associations. Because network statistics are often used to describe the structure of a network and our motivation for this analysis was not to describe the structure of the microbial networks at SPOT, we chose to refrain from presenting network statistics in the manuscript.

Line 429: Remove "the" in "...the algal endosymbionts".

We feel that the word 'the' is appropriate in this context.

Lines 431: The headings in this section could be more direct. Something like, "Diatoms involved in mutualistic epibioses."

We agree that we could have made the section header more direct. We have revised this as suggested (line 643).

Line 444: Remove “the” in “...the MAST-3 cells...”.

We have revised our wording accordingly (line 660).

Lines 447-449: These are cool findings, but I wonder if they could also be heterotrophic relationships at times. How can you be certain that these protist grazers are not consuming these prey items? I would include another sentence to clarify that this may also reflect grazing.

While MAST cells are heterotrophic, they are relatively small ($< 5 \mu\text{m}$) and are therefore not in the right size class to be grazing on diatoms. Available information relating to MAST cells indicate that they are very small, important bacterial grazers in oceanic environments (del Campo et al. 2013, Massana et al. 2014).

Lines 451-453: This sentence seems contradictory. You say many of them were identified but then mention that it was only ~2%. I think it is ok to say that only 2% of the associations were revealed in PIDA at the level that you could resolve in your amplicon data; however, your approach allowed for identification of putative association that may reflect actual interactions and spur additional studies to verify them. The point is that most interactions in situ are unresolved, and so, more studies like yours will be important to guide future efforts.

We see how this language may be contradictory, and we have revised this sentence (line 679).

Line 455: Can you deal with false positives in NetGAM? I know that programs like Spiec-Easi aim to mitigate false positives and indirect relationships.

The NetGAM transformation and the graphical lasso networking technique (i.e., the technique that is leveraged in Spiec-Easi) that we used for our network analyses should prevent us from capturing “relationships” that occur due to temporal dependence or indirect interactions. We included this discussion on false positives in our manuscript because even when leveraging these methods, the networks may still recover some false positive interactions. We feel that it is important to be mindful that the potential for false positives exists when evaluating network predictions.

Line 463: What do you mean by “real”?

The word “real” here was used to differentiate true positive network predictions from false positive network predictions. We realize that this was unclear and have revised this sentence (line 694-696).

Line 485: What is meant by strong? I would be quantitative if possible.

We revised this section to be more quantitative (lines 738-788) and added the SCC and adjusted p -value of these strong interactions to Table S7.

Figure 1: I think it is fine that your taxonomic groupings do not follow a specific level in PR2, but I would clarify in the legend or methods that you grouped them manually into these categories. For instance, all diatoms and MAST were grouped but you separated out Syndiniales into the two major groups. Do the “other” groupings reflect a certain taxonomic level and relative abundance cutoff? Why keep metazoans in the analysis, if you were trying to filter them out in the first place? I would remove them, as the paper is focused on protist-protist associations. Shouldn't the number of samples in each month be 14? One for each year of the dataset? Oh, but some samples were removed. I would clarify in the legend.

We thank the reviewer for pointing out that our taxonomic groupings are unclear. We have revised the figure legend to clarify how we manually grouped our ASVs. We have also added additional information about the months of sampling that were missed during the 14 year and 9-month time-series to the main text to provide context for why the number of samples in each month is less than 14.

Figure 3: I like this figure, but it is a bit small and hard to read. Can you increase the figure size and make the data points and vectors larger? The shapes and colors distinguish month and season, correct? It took me a minute to figure that out. What factors were significant to the RDA? Did this vary between surface and DCM? I would clarify in the legend and mention in the paper somewhere what factors (if any) were found to not be significant in the stepwise selection.

We increased the size of the points, vectors, and words in Figure 3 to make it more legible and added text to the figure caption to provide additional information about what the different colors/shapes mean. We also added information to the caption about which factors were significant in driving differences in community composition at the surface and DCM.

Figure 4: I like the idea, but this figure is hard to interpret. The nodes are way too small, and so, we cannot see the colors and taxonomic groups that they represent. Is there another way to summarize these connections? Or maybe move this to supplemental and focus on the top 10-20 connections? You could also include a venn diagram to show the overlap between SPOT and PIDA parasitic associations. The connections in panels a and c are easier to interpret if you increased the figure size. However, panels b and d are hard to interpret with all the gray lines present. Maybe try a chord type plot and color based on the type of connection. This may help to avoid all the single connecting lines.

Figure 5: This plot is larger and easier to read, though it does look a bit warped to me. I would suggest trying out chord plots that may improve the figure and make it easier to read. I appreciate the colors and letters to distinguish node, but it takes a long time staring at the figure to interpret what is happening. I would try a chord plot and see if that makes it any clearer.

We thank the reviewer for pointing out that Figures 4 and 5 are hard to interpret. We originally chose to display the interactions in this way to visualize specific ASV-ASV interactions; however, we agree that visualizing ASV-ASV interactions may not be the best approach when there are hundreds of interactions. Therefore, we have replaced Figures 4 and 5 with chord diagrams as the reviewer suggested and moved the ASV-ASV interaction diagrams to the supplemental materials (Figure S4 and S5).

Figure 7: All spearman values in this figure were significant, right? I would give the ASV number as well in case some of the same ASVs were found twice or they were unique but had the same taxonomic assignment. Maybe that situation never occurred?

Yes, the p -values of the correlation analyses were significant. We have recorded the SCCs that resulted from these analyses and the associated p -values in a supplemental table (Table S7).

Re: mSystems01045-24R1 (With a little help from my friends: Importance of protist-protist interactions in structuring marine protistan communities in the San Pedro Channel)

Dear Dr. Samantha Gleich:

As an editor it is nice to comments like these from the reviews.

Your manuscript has been accepted, and I am forwarding it to the ASM production staff for publication. Your paper will first be checked to make sure all elements meet the technical requirements. ASM staff will contact you if anything needs to be revised before copyediting and production can begin. Otherwise, you will be notified when your proofs are ready to be viewed.

Sincerely,
Jeffrey Blanchard
Editor
mSystems

Reviewer #1 (Comments for the Author):

I appreciate the additional analyses and revisions made to the manuscript, which have further clarified protist-protist interactions and addressed some limitations related to the lack of accumulated knowledge. I thoroughly enjoyed the current version of the manuscript and have no additional comments.

Reviewer #2 (Comments for the Author):

I want to thank the authors for responding to my reviews and taking into consideration most of my suggestions and comments. Thank you for your time and effort on this manuscript.